# Electro-Mineralization of Aqueous Phenazopyridine Using Platinum Nanoparticles Deposited onto Multiwalled Carbon Nanotubes

Ibrahim M. Nassar [1], Heba Nassar [1], Mohyeddin Assali [2], Muath H. S. Helal [3], Hyobin Han [4], Tae Woo Kim [4,*], Mazen Salman [5] and Hikmat S. Hilal [1,*]

1 Department of Chemistry, College of Science, An-Najah National University, Nablus P400, Palestine; i.nassar@najah.edu (I.M.N.); h.nassar@najah.edu (H.N.)
2 Department of Pharmacy, Faculty of Medicine and Health Sciences, An-Najah National University, Nablus P400, Palestine; m.d.assali@najah.edu
3 Department of Biomedical Sciences, Faculty of Medicine and Health Sciences, An-Najah National University, Nablus P400, Palestine; muath.helal@najah.edu
4 Hydrogen Research Department, Korea Institute of Energy Research, 152, Gajeong-ro, Yuseong-gu, Daejeon 34129, Republic of Korea; hyobin@kier.re.kr
5 Department of Agricultural Biotechnology, Palestine Technical University—Kadoorie, Tulkarm P305, Palestine; m.salman@ptuk.edu.ps
* Correspondence: twkim2015@kier.re.kr (T.W.K.); hshilal@najah.edu (H.S.H.)

**Abstract:** Water may easily become polluted by pharmaceutical wastes, such as phenazopyridine hydrochloride. The pollutant can be removed by electrochemical oxidation in the form of minerals. A novel electrode has been developed for this purpose. Pt nanoparticles (PtNPs) are electrodeposited onto multiwalled carbon nanotubes supported onto fluorine-doped tin oxide (FTO/Glass). The resulting PtNP@MWCNT-FTO-E electrode is characterized by X-ray diffraction, atomic force microscopy, scanning electron microscopy, electron diffraction spectroscopy and X-ray photoelectron spectroscopy. The electrode exhibits high efficiency in the electrochemical oxidation process thanks to the large specific surface area of the PtNPs and their ability to behave as charge transfer catalysts. The contaminant undergoes complete mineralization, leaving no organics after treatment. The resulting nitrate ions further confirm contaminant mineralization, but fortunately, they disappear over time, which confirms the safety of the process in water treatment. Moreover, the electrode operates under a variety of applied potentials, pH values, temperatures and contaminant concentrations. The electrode exhibits high stability upon recovery and reuse while retaining its physical characteristics before and after use. This study highlights the benefit of using Pt nanoparticles in the electro-degradation of aqueous organic contaminants, especially waste pharmaceuticals, for the first time. It also recommends scaling up the process and studying the continuous-flow reaction process to assess the economic and technical feasibility in future large-scale applications.

**Keywords:** Pt nanoparticles; multiwalled carbon nanotubes; fluorine-doped tin oxide; complete electro-mineralization of phenazopyridine; electrode stability and recovery



## 1. Introduction

Pure natural water is becoming scarce for many reasons, including pollution with various species, including organic pharmaceuticals [1–7]. Both surface and ground waters can be contaminated [8,9]. Reports show that water and aqueous suspended particles may reach up to ~82 mg/kg [10]. Phenazopyridine is a drug that is widely used as a urinary analgesic to treat urinary tract infections. Unfortunately, about 50–60% of up-taken phenazopyridine is secreted through urination after use by humans [11,12]. This leads to environmental and water pollution from hospital and industrial effluents, where both surface and ground waters are being reached [13–15]. Despite its low solubility in water,

less than 100 mg/L at neutral conditions [16], phenazopyridine contaminant is hazardous and should be seriously considered [16].

Water reclamation from various organic contaminants, including phenazopyridine, is imperative. Various treatment methods are known, including adsorption, membrane separation and others [17–21]. New methods for remediation are being developed [22–24]. Photocatalytic degradation of phenazopyridine was described [25–27]. Phenazopyridine was degraded by ozone oxidation using iron compounds [28].

Aqueous phenazopyridine removal by combined biological and electrochemical methods via the electro-Fenton process was described [13]. Electrochemical oxidation of phenazopyridine was studied using multi-layered $IrO_2$-$Ta_2O_5$ anodes coated on Ti substrates [29], where up to 72% was degraded in 180 min. The electrode efficiency varied for catalyst composition and solution pH. Sadikuglu et al. studied the irreversible electrooxidation of phenazopyridine using conventional ultra-trace graphite electrodes, revealing that the oxidation process leads to dimerization of the phenazopyridine [30], not complete mineralization or removal. In another study, Sadikuglu et al. reported the use of a glassy electrode modified with an electro-polymerized p-aminobenzene sulfonic acid film for the detection of phenazopyridine HCl [31]. Fe-doped ZIF-8 zeolite was used as an electrode in the galvanostatic electrooxidation of phenazopyridine HCl, with pH 7 being the optimal pH for the reaction progression [32]. The process was effective only at low phenazopyridine concentrations in the range of 10–30 mg/L but not in highly contaminated waters. Magnetite nanoscale-activated carbon electrodes were used to degrade phenazopyridine in water by the electro-Fenton process using the galvanostatic mode at a constant current. The study showed the possibility of effectively removing 98% of the contaminant at optimal conditions and a low pH value of 3 [33].

2-aminopyridine was electrochemically degraded using Pt electrodes [34]. A Pt anode was used to electro-degrade various organic contaminants, such as synthetic dyes [35], phenols [36,37], glucose [38], herbicides [39], chlorophenols [40] and methanol [41]. In one report, phenazopyridine electrochemical detection was performed on Pt electrodes [42]. However, reports describing the removal of drugs and phenazopyridine from water using Pt-based electrodes, let alone nanoscale Pt (PtNP), are not available.

In this communication, safe, durable and economic processes to remove phenazopyridine from water are targeted. While using Pt sheet electrodes can be useful, their high cost is a limitation. In order to save electrode costs, nanoparticles of Pt will be assessed here. PtNPs are widely considered in basic and applied research for various applications [43,44]. In the nanoscale, small amounts of Pt are used rather than bulk amounts. This lowers the material cost. The PtNPs have a higher specific surface area (SSA), which increases the process efficiency and lowers the process costs. Moreover, the recovery and reuse of the PtNPs are critically important for more than one reason. Firstly, they prevent environmental contamination by the process outcomes. Secondly, they save material cost by multiple reuse.

This study is based on a number of assumptions. First, Pt nanoparticles have high electrical conductivity and can effectively transfer electrical charge at the solid electrode/solution interface. Second, Pt nanoparticles have a much higher specific surface area (SSA) than Pt sheets. This would speed up the charge transfer across the electrode/solution interface. Third, Pt nanoparticles are stable under various conditions, which means that the electrode would be recovered and reused multiple times, as described above. Fourth, supporting the Pt nanoparticles on other high SSA-conducting solid matrices, such as multiwalled carbon nanotubes (MWCNTs), should allow the particles to disperse onto the mesh rather than forming a compact Pt film. This should allow contaminant molecules to penetrate through the mesh and reach the dispersed Pt particles, consequently speeding up the electrical charge transfer.

All these assumptions will be tested here. Fluorine-doped tin oxide on oxide-glass (FTO/Glass) substrates will be employed as beds. The beds will be used as substrates to deposit Pt nanoparticles, which are expected to form Pt films. In parallel with that, FTO

beds will be used as support for MWCNT matrices. The MWCNT-FTO beds will then be used to support Pt nanoparticles by various methods, where the resulting Pt is expected to exist as dispersed particles on MWCNTs rather than compact films. All electrodes will be characterized before being comparatively assessed in the electrochemical mineralization of contaminant phenazopyridine. To our knowledge, no similar studies were reported earlier.

## 2. Experimental Section

### 2.1. Starting Materials

All starting materials are of chemically pure grade. Platinum foils (99.95% pure, 2 cm × 2 cm × 0.2 mm) are purchased from Sigma-Aldrich local venders in Nablus, Palestine. Multiwalled carbon nanotubes (functionalized with carboxylic acid) are purchased from Sigma-Aldrich, (Model No. 755125). $H_2PtCl_6$ solution (8% $w/w$) is purchased from Sigma-Aldrich (Model No. 262587). The phenazopyridine hydrochloride, structure shown in Scheme 1 below, is kindly donated by Birzeit-Palestine Pharmaceutical Company. FTO/Glass slides from Sigma-Aldrich (Model No. 735167, with surface resistivity of ~7 Ω/sq) are cut into smaller slides (2 cm × 5 cm). Other common solvents and salts are from Sigma-Aldrich.

**Scheme 1.** Phenazopyridine HCl structure.

### 2.2. Equipment

A Corrtest Potentiostat (Cs 350) is used as a power source. Solid sample heating is performed on a Boekel 107905 furnace. Atomic absorption spectrometry is measured on a ThermoFisher Scientific-iCE 3000 spectrometer (Grand Island, NY, USA). A Power sonic 405 sonicator is used.

A high-performance liquid chromatogram (Waters 1525) with a photodiode-array UV–Visible detector and auto-sampler injector at a detection wavelength of 280 nm is employed to measure the phenazopyridine remaining in solution together with other possible organic by-products. The eluent solution is 25% water and 75% acetonitrile with a 1.0 mL/min flow rate and 20 μL injection volume. Phenazopyridine spectra are recorded on a Shimadzu (UV-1800) spectrophotometer. In order to check for organic matter, total organic carbon (TOC) is measured over time. A SHIMADZU analyzer (Model TOCL CSH/CSN) at PTU, Tulkarem, is used for this purpose.

Atomic force microscopy (AFM) is measured on a CoreAFM manufactured by Nanosurf, Switzerland. The AFM image is created and analyzed by Gwyddion software (http://gwyddion.net/).

A JEOL (JSM-6700F) Field Emission SEM is used to measure scanning electron micrographs. A PAN-alytical X'Pert-PRO X-ray diffractometer with a Cu Kα source is used to measure X-ray diffraction (XRD) measurements. X-ray photoelectron spectra (XPS) are measured on a Multi-Lab 2000 spectrometer with a micro-focusing mono-chromated Al-Kα X-ray (1486.6 eV) source. All XRD, XPS, SEM and EDS are measured at the Korean Institute of Energy Research (Daejeon, Republic of Korea).

### 2.3. Electrode Preparation

Each preparation is repeated 3 times to check reproducibility. The following electrodes are prepared.

- Platinum sheet electrode

Platinum sheet electrodes (2 cm $\times$ 2 cm x 0.2 mm, 1.71 g, 0.088 mole) are cleaned by rinsing several times with distilled water and firing for a few seconds before use. The electrodes are termed Pt electrodes.

- FTO Electrode

Before usage, the FTO/Glass slides are sonicated inside isopropyl alcohol and inside acetone (for 5 min each time). Then, they are rinsed many times with deionized water. In each case, the used area is $2 \times 4$ cm$^2$, while the other area is protected by coating with aluminum foil. The electrodes are called FTO electrodes, with an approximate surface area of less than 12 m$^2$/g FTO ($\pm 15\%$) based on the acetic absorption method [45,46].

- Nano Pt@FTO/Glass electrodes

FTO/Glass substrates are coated with Pt nanoparticles by room-temperature electrodeposition. Deposition is conducted on the Corrtest electrochemical workstation, a three-electrode 100 mL cell, with a Pt sheet counter electrode, an SCE reference electrode and the FTO substrate as a working electrode. Various routes are assessed to produce a stable PtNP@FTO/Glass electrode. Prior to use, the FTO/Glass substrates are cleaned as described above.

(a) Thermal spray deposition. H$_2$PtCl$_6$ solution (0.005 M) is prepared by diluting 2.0 mL of stock H$_2$PtCl$_6$ solution (8% $w/w$, 0.205 M) in 39 mL absolute ethanol to yield a solution of 0.01 M. The solution is sonicated for (5 min), placed in a sprayer and sprayed on FTO substrate surfaces at 80 °C for 20 min at a fixed distance. Deposition is performed on 4 FTO substrates, each with a net area (2 cm $\times$ 4 cm), where the other part (2 cm $\times$ 1 cm) is protected by coating with aluminum foil. The films are left for 24 h to dry, after which they are sintered at 350 °C for 15 min. The electrode, termed PtNP@FTO-T, contains $2.0 \times 10^{-2}$ g ($1.0 \times 10^{-4}$ mol) Pt distributed over the 8 cm$^2$ area. The approximate specific surface area is 50 ($\pm 15\%$) m$^2$/g.

(b) Electrochemical deposition. The literature [47–49] shows that H$_2$PtCl$_6$ can be reduced to metal Pt nanoparticles onto various substrates with special characteristics. The methods are followed here to deposit Pt nanoparticles onto 10 FTO/Glass substrates. Electrodeposition is performed at $-0.80$ V (vs. SCE) for 300 sec from a stirred solution of H$_2$PtCl$_6$ (4.0 mL of 0.205 M) and KCl (78 mL, 0.10 M). The resulting electrodes are then rinsed with deionized water before drying (room temperature) and storage. The resulting electrode is termed PtNP@FTO-E. The electrode contains $1.5 \times 10^{-2}$ g, $7.7 \times 10^{-5}$ mole Pt distributed over 8 cm$^2$. Based on earlier reports [50], the approximate specific surface area is 90 m$^2$/g.

- MWCNT-FTO/Glass electrodes

MWCNTs, functionalized with COOH groups, are deposited onto FTO/Glass substrates. The functionalized MWCNTs are intentionally used to improve attachment with the FTO surface, as described earlier [51].

Coating the FTO electrode surface with MWCNTs follows earlier literature procedures [52–54]. A spraying suspension is prepared by mixing MWCNTs (1.0 mg) with anhydrous ethanol (20 mL). After 15 min sonication in a bath, the suspension is sprayed on the FTO substrates and heated at 80 °C for 20 min at a fixed distance. Deposition is restricted to an area of after (2 cm $\times$ 4 cm) by coating the other part with aluminum foil. The modified electrode is named MWCNT-FTO. The electrode contains 0.1 mg MWCNTs distributed over an area of 8 cm$^2$. Based on vendors and the literature, the approximate surface area for the MWCNTs is 300 m$^2$/g [55].

- MWCNT-FTO electrode coated with Pt nanoparticles

Platinum nanoparticles are electrodeposited on 16 MWCNT-FTO electrodes at $25 \pm 1$ °C from a deposition solution of H$_2$PtCl$_6$ (4.0 mL of 0.205 M) and KCl (78 mL, 0.1 M). The deposition is performed at $-0.80$ V two times for 300 s. The resulting electrode is rinsed

with deionized water, air-dried and stored. The electrode is termed PtNP@MWCNT-FTO-E. The Pt content in the electrode is $1.0 \times 10^{-2}$ g ($5.1 \times 10^{-5}$ mol) Pt in 0.1 mg MWCNTs, distributed over an area of 8 cm$^2$. The overall specific surface area is less than 100 m$^2$/g for composite Pt@MWCNT, based on the literature [56,57].

*2.4. Electrode Characterization*

The prepared electrodes PtNP@FTO-T, PtNP@FTO-E, MWCNT-FTO and PtNP@MWCNT-FTO-E are characterized by scanning electronic microscopy (SEM), X-ray diffraction (XRD), energy-dispersive X-ray spectra (EDS) and X-ray photoelectron spectroscopy (XPS). The electrode PtNP@MWCNT-FTO-E is characterized in its fresh form and after recovery from an electrochemical experimental study.

The Pt nanoparticle sizes are calculated from XRD patterns using the Scherrer equation (Equation (1)):

$$S = 0.9\lambda / \beta \cos\theta \tag{1}$$

where $S$ denotes average particle size (in Å), $\lambda$ wavelength for Cu K$\alpha$ radiation (1.5406 Å), $\beta$ is full width at half maximum (FWHM) in radians and $\theta$ is diffraction angle (degree) [58]. Bragg's law (Equation (2)) helps determine interplanar distances [59,60]:

$$2d \sin\theta = n\lambda \tag{2}$$

where $d$ denotes interplanar spacing in the crystal, n is an integer and $\theta$ is the incident angle.

*2.5. Phenazopyridine Electro-Degradation Experiments*

Phenazopyridine electrooxidation experiments are performed in a magnetically stirred three-electrode glass cell (100 mL) connected to the Corrtest workstation. Typically, solutions of 40 ppm phenazopyridine are used, with an intrinsic pH of ~3.0. Other experiments are conducted at various pH values, as described below. Control experiments show that stirring improves contaminant removal; thus, the reaction mixture is magnetically stirred. The reference electrode is SCE, the counter electrode is an FTO plate and the working electrode is one of the prepared electrodes. The distance between the counter and working electrodes is set at 1.2 cm, as larger distances (e.g., 3.2 cm) yield less contaminant removal. Unless otherwise stated, the geometrical exposed area of each working cathode is 7.80 cm$^2$.

At specific time intervals, aliquots are syringed out from the reaction solution, diluted properly and spectrophotometrically analyzed for remaining phenazopyridine concentration at 430 nm, as described earlier [61]. A calibration curve (Supplementary Figure S1) is constructed for this purpose.

All prepared electrodes are tested in the electro-degradation of phenazopyridine in order to find the one with the highest efficiency. Each electrode is prepared multiple times, and each prepared electrode is used three times in the electrooxidation reaction. Various reaction parameters, such as pH, temperature, reactant concentration and others, are examined to determine the process's applicability under various conditions.

*2.6. Statistical Analysis*

All electro-degradation experiments are repeated three times each, and the measurements are in triplicates. Statistical analysis is performed using Xlstat (version 2021) or Excel (Adinosoft 2021) [62]. Data with different letters are significantly different after Tukey's HSD test using ANOVA at $p < 0.05$. Bars describe standard error (SE) values.

## 3. Results and Discussion

*3.1. Electrode Characterization*

- XRD:

The XRD patterns for various electrodes are shown in Figure 1. The figure shows patterns for PtNP@FTO-T and PtNP@FTO-E electrodes. In PtNP@FTO-T and FTO/Pt-E, the signals observed at 2θ = 39.96° (111), 46.68° (200), 68.54° (220) and 82.22° (311) are due

to face-centered cubic (fcc) Pt (JCPDS Card 04-0802) [63]. The patterns thus confirm the Pt deposition on the FTO surface. The other signals at 2θ = 26.48, 33.70, 37.74, 51.48, 54.45, 61.68, 65.68 and 78.7°, associated with (110), (101), (200), (211), (220), (310), (301) and (321) planes, respectively, are due to the FTO layer (JCPDS, 41-1445) [64].

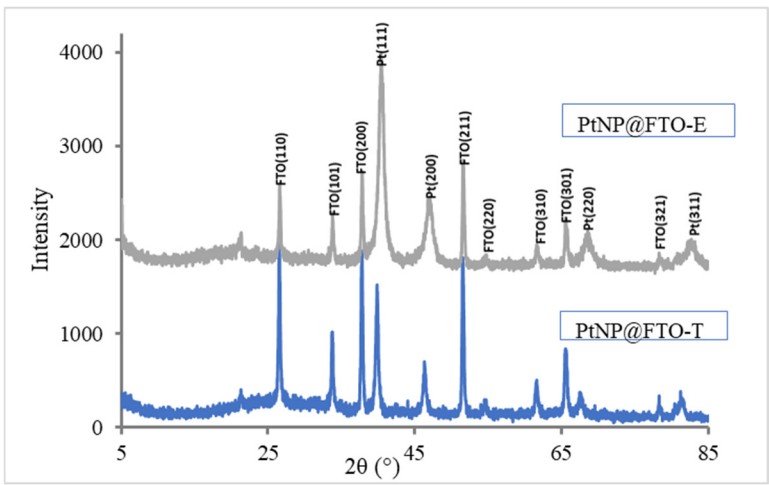

**Figure 1.** XRD patterns measured for PtNP@FTO-T and PtNP@FTO-E electrodes.

The wide signals, with low height, observed for the Pt signals are an indication of nanoscale crystallites, even if they may exist in films, as described below. In the literature, nanoparticles typically have wide XRD patterns with lower heights, even when they exist in films, as widely described in the literature [65,66]. Table 1 summarizes XRD approximate results for the two electrodes, including crystallite sizes and interplanar distances. The electrode PtNP@FTO-T has a smaller Pt particle size than the PtNP@FTO-E electrode.

**Table 1.** Summary of XRD patterns of PtNP@FTO-E and PtNP@FTO-T electrodes.

| Electrode | 2θ ° | FWHM | Species | Particle Size * (nm) | Average Particle Size * (nm) | d (Å) | (*hkl*) |
|---|---|---|---|---|---|---|---|
| PtNP@FTO-E | 39.80 | 0.44 | Pt | 20.06 | | 2.8 | (111) |
| | 46.33 | 0.22 | Pt | 41.04 | 38.60 | 2.0 | (200) |
| | 67.62 | 0.14 | Pt | 71.37 | | 1.4 | (220) |
| | 81.84 | 0.22 | Pt | 49.94 | | 1.2 | (311) |
| PtNP@FTO-T | 40.22 | 0.62 | Pt | 14.26 | | 2.2 | (111) |
| | 46.68 | 0.40 | Pt | 22.60 | 21.65 | 1.9 | (200) |
| | 68.18 | 0.45 | Pt | 22.28 | | 1.4 | (220) |
| | 82.23 | 0.50 | Pt | 22.04 | | 1.2 | (311) |

* Measurement error is ±10%.

XRD patterns of MWCNT-FTO and PtNP@MWCNT-FTO-E electrodes are shown in Figure 2. The latter electrode is measured before and after use in an electrooxidation reaction. The signals at 2θ = ~27, 33.70, 37.74, 51.48, 54.45, 61.68, 65.68 and 78.7° belong to the FTO substrate (JCPDS, 41-1445) [64,67]. MWCNT reflection observed at ~24° is due to (002), as reported earlier [68–70] (JCPDS, 01-0646).

The fresh and used PtNP@MWCNT-FTO-E electrodes show signals due to Pt at 2θ = 39.96° (111), 46.68° (200), 68.54° (220) and 82.22° (311), as reported earlier (JCPDS Card 04-0802) [63]. This confirms the occurrence of Pt in the electrodes.

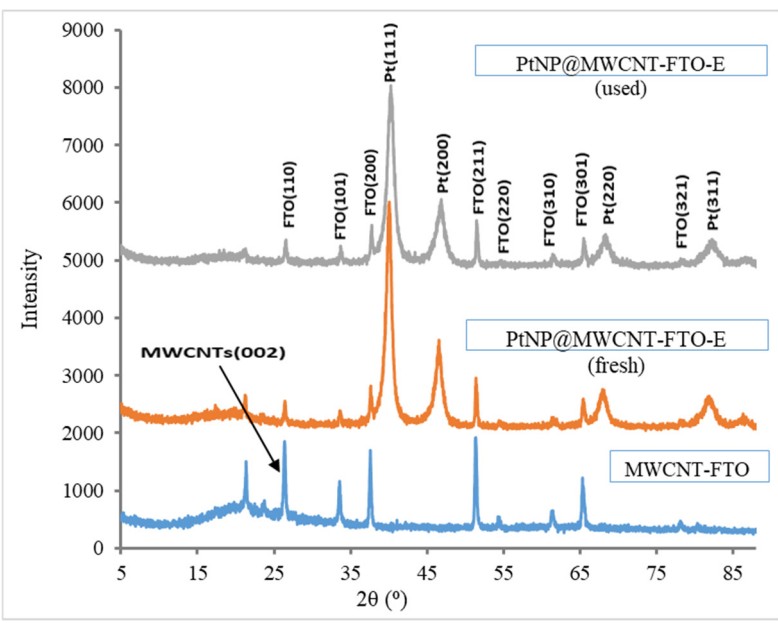

**Figure 2.** XRD patterns measured for MWCNT-FTO, PtNP@MWCNT-FTO-E (fresh) and PtNP@MWCNT-FTO-E (used) electrodes.

In Figure 2, with deposited Pt nanoparticles, the FTO and MWCNT signals are weaker than for the PtNP@MWCNT-FTO-E electrode. Moreover, no significant effect on the PtNP@MWCNT-FTO-E is observed after recovery and reuse. This indicates the electrode stability on reuse due to platinum stability to oxidation, vide infra, [6]. Table 2 summarizes the RXD approximate data for all three electrodes. As the particle size measurements are only approximate, the Pt nanoparticles retain their sizes in the prepared electrode after usage in the electrooxidation process.

**Table 2.** Summary of XRD for MWCNT-FTO and PtNP@MWCNT-FTO-E electrodes (fresh and used).

| Electrode | 2θ (°) | FWHM | Species | Particle Size (nm) * | Average Particle Size (nm) * | d (Å) | (*hkl*) |
|---|---|---|---|---|---|---|---|
| MWCNTs-FTO | 26.40 | 0.26 | MWCNTs | 32.79 | 32.79 | 3.4 | (002) |
| PtNP@MWCNT-FTO-E (fresh) | 26.40 | 0.24 | MWCNTs | 35.52 | | 3.4 | (002) |
| | 39.80 | 0.44 | Pt | 20.06 | | 2.8 | (111) |
| | 46.33 | 0.22 | Pt | 41.04 | 45.60 | 2.0 | (200) |
| | 67.62 | 0.14 | Pt | 71.37 | | 1.4 | (220) |
| | 81.84 | 0.22 | Pt | 49.94 | | 1.2 | (311) |
| PtNP@MWCNT-FTO-E (used) | 26.40 | 0.18 | MWCNTs | 47.37 | | 3.4 | (002) |
| | 40.22 | 0.62 | Pt | 16.26 | | 2.2 | (111) |
| | 46.68 | 0.40 | Pt | 36.60 | 40.11 | 1.9 | (200) |
| | 68.18 | 0.45 | Pt | 55.28 | | 1.4 | (220) |
| | 82.23 | 0.50 | Pt | 45.04 | | 1.2 | (311) |

* Measurement error is ±10%.

- Surface morphology

Figure 3 shows SEM images for PtNP@FTO-T and PtNP@FTO-E electrodes. Both surface and cross-sectional images are shown. Both electrodes involve films of Pt nanoparticles at the surface. The thermally deposited film involves particles that are densely arranged on the FTO surface and interconnected. This is due to the sintering resulting from relatively high-temperature treatment. In the PtNP@FTO-T electrode, the Pt film thickness is ~1.2 μm (±10%), with a total mass of $2.0 \times 10^{-2}$ g. On the other hand, the electrodeposited electrode involves films with flower-like particles that are less densely arranged on the

FTO surface. The average thickness values of the Pt layer deposited in the PtNP@FTO-E electrode, calculated based on cross-sectional SEM images, is ~2.5 μm (±10%) with a total mass of ~$1.5 \times 10^{-2}$ g.

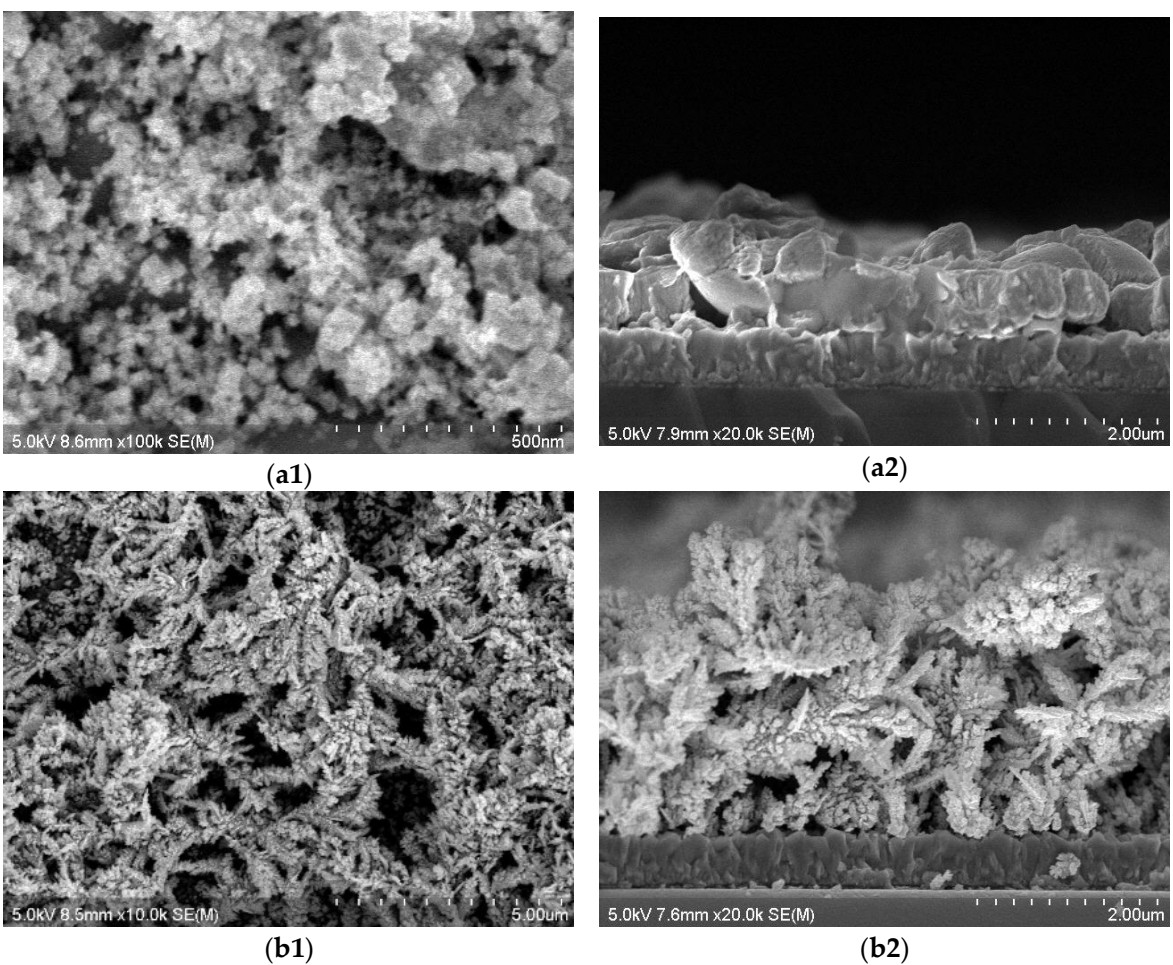

**Figure 3.** SEM micrographs for (**a**) PtNP@FTO-T and (**b**) PtNP@FTO-E electrodes. Subfigures (**a1**,**b1**) are for surface micrographs, while subfigures (**a2**,**b2**) are for cross-sectional micrographs.

In both films, relatively large particles are observed. These observations should not be confused with earlier XRD results discussed above. While the XRD results show nanoscale crystallites with small sizes, the SEM micrographs show larger agglomerates in which the crystallites reside.

Figure 4 shows surface and cross-sectional SEM images for MWCNT-FTO, together with fresh and used PtNP@MWCNT-FTO-E electrodes. The micrographs indicate that the Pt, electrodeposited on the MWCNT-FTO surface, involves randomly distributed Pt particles coating the surface of the tubes. Mesh-like structures can be observed for MWCNT-FTO and PtNP@MWCNT-FTO-E electrodes. Unlike the PtNP@FTO-E electrode, no Pt films can be observed in the PtNP@MWCNT-FTO-E electrodes. The Pt crystallites are too small, and their sizes cannot be accurately determined by the SEM. The crystallites cover the surfaces of the individual tubes but do not form a continuous thin film all over the electrode surface. However, the existence of the Pt in the PtNP@MWCNT-FTO-E electrodes is confirmed by the XRD patterns above.

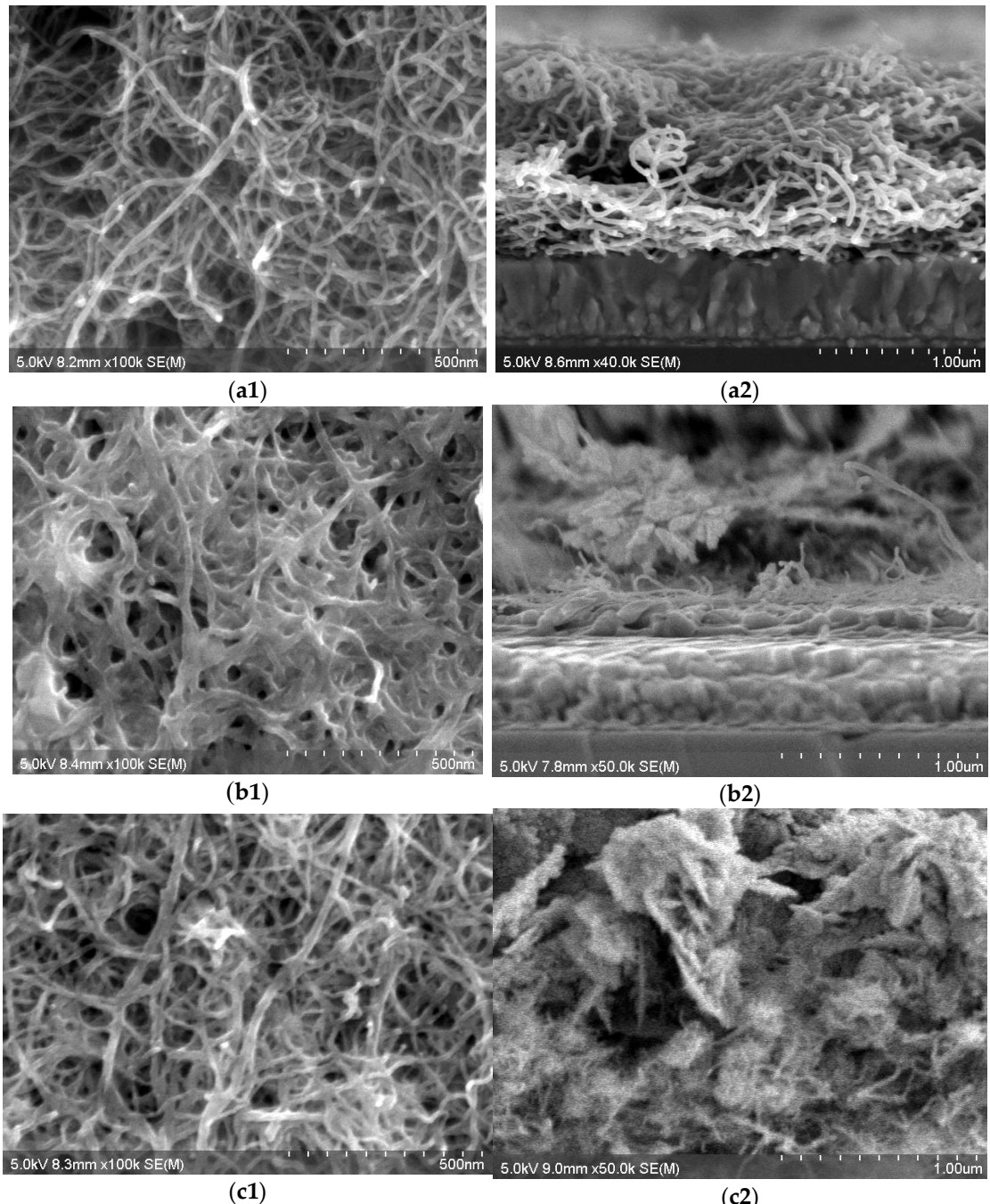

**Figure 4.** Surface and cross-sectional micrographs measured for (**a**) MWCNT-FTO, (**b**) fresh PtNP@MWCNT-FTO-E and (**c**) used PtNP@MWCNT-FTO-E electrodes. Subfigures (**a1,b1,c1**) are for surface micrographs, while subfigures (**a2,b2,c2**) are for cross-sectional micrographs.

SEM further confirms the occurrence of nanoparticles on the PtNP@MWCNT-FTO-E electrodes. In the MWCNT-FTO electrode, the SEM shows that the tube is only ~12 nm in diameter. PtNP@MWCNT-FTO-E has tubes of a diameter of 22 nm or more. The difference is due to the Pt nanoparticles residing on the MWCNTs (Figure 4). Surface and cross-sectional micrographs are measured for (a) MWCNT-FTO, (b) fresh PtNP@MWCNT-FTO-E and (c) used PtNP@MWCNT-FTO-E electrodes. Figure 4(a2,b2,c2) describe the cross-sectional micrographs.

Film thicknesses are measured from cross-sectional images. The approximate values are more than 2.5 μm for MWCNT-FTO, fresh PtNP@MWCNT-FTO-E and used fresh PtNP@MWCNT-FTO-E, respectively, with 10% error.

To further confirm the presence of dispersed Pt particles in the PtNP@MWCNT-FTO-E electrode, the atomic force microscopy (AFM) is measured for the electrode. The micrograph (Figure 5) confirms the presence of Pt particles dispersed at the MWCNT surface. Figure 5A shows the formation of the spherical Pt particles with an average diameter of 200 nm, as depicted by arrows. The particles are agglomerates of Pt, and each involves smaller nano-crystallites, as confirmed above by the XRD patterns. Figure 5B shows the MWCNT layer deposited on the FTO substrate. The MWCNT layer has an AFM that resembles earlier literature micrographs [71,72].

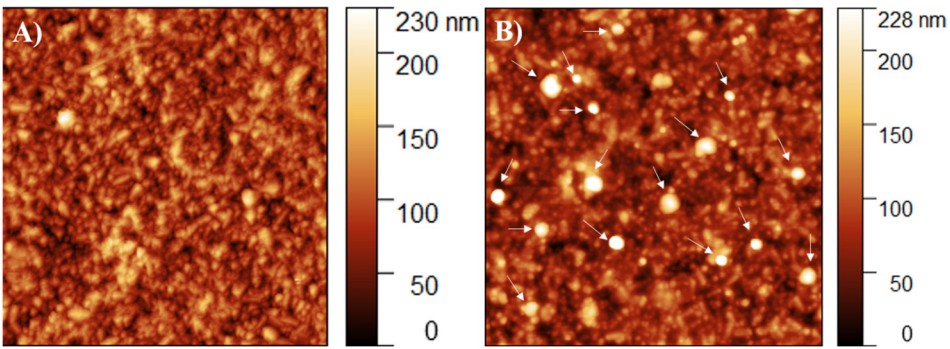

**Figure 5.** AFM image measured for the Pt particles and MWCNTs. (**A**) For MWCNT-FTO only, and (**B**) for PtNP@MWCNT-FTO-E electrode. The arrows in (**B**) show the spherical Pt particles (agglomerates) dispersed onto the MWCNT surface.

Due to its special efficiency in electrooxidation reactions, the fresh and used PtNP@MWCNT-FTO-E electrodes are analyzed for elements using the EDS method (Figure S2). That is to further confirm the presence of Pt. Both electrodes involve Pt as a main element at the surface. The EDS results are normally considered only approximate ones, but they still provide evidence about the occurrence of Pt on the electrode surface, as summarized in Table 3. The high Pt content is due to the presence of Pt crystallites on individual MWCNT surfaces.

**Table 3.** Summary of elemental analysis for fresh and used PtNP@MWCNT-FTO-E electrode based on EDS spectra. Measurement error ~15%.

| Electrode | Element | Atomic% | Mass% |
|-----------|---------|---------|-------|
| Fresh | C K | 37.4 | 3.3 |
| | Pt M | 56.7 | 88.4 |
| | Sn L | 8.7 | 8.3 |
| Used | C K | 34.6 | 3.3 |
| | Pt M | 56.7 | 88.4 |
| | Sn L | 8.7 | 8.3 |

More evidence of the occurrence of Pt at the PtNP@MWCNT-FTO-E electrode is observed from XPS spectra in Figure 6. Figure 6a shows measured complete spectra for fresh and used systems. Figure 6b,c show separate peaks for separate elements in fresh and used electrodes, respectively, for confirmation. The Pt and C elements can be easily detected. The fresh and used electrodes exhibit no significant change in Pt spectra, which indicates electrode stability in the electrooxidation process.

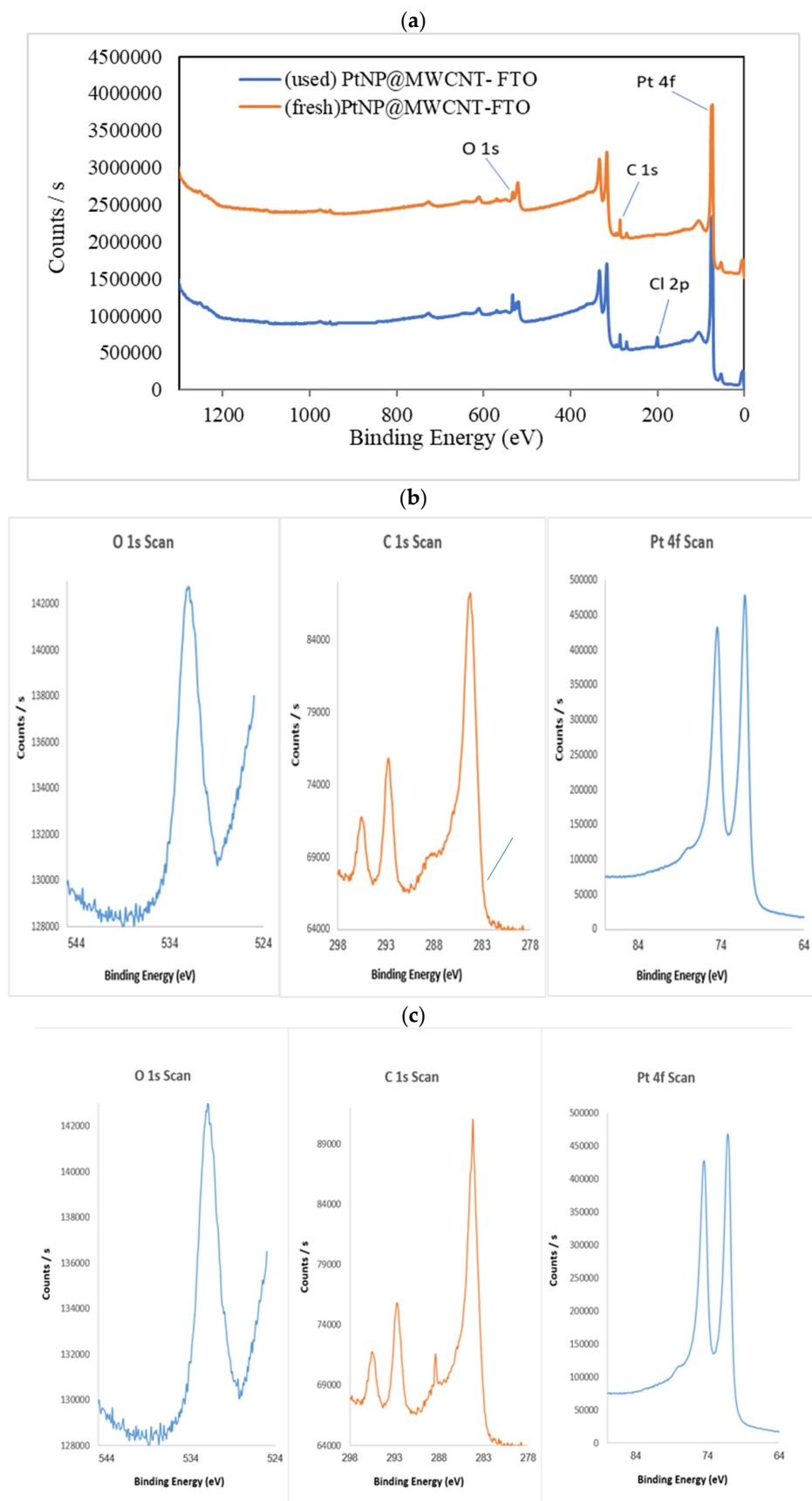

**Figure 6.** XPS spectra measured for fresh and used PtNP@MWCNT-FTO-E electrodes. (**a**) Complete spectra are shown; (**b**,**c**) spectra for separate ions in fresh and used electrodes.

The C 1s peak at 286.03 eV is due to MWCNTs [73]. For C 1s, the peaks located at 284.5 eV and 284.8 eV are due to C=C bonds with ($sp^2$) and C-C bonds ($sp^3$). The 285.7 eV peak is due to both C=O and C-OH groups. The 289.4 eV peak is due to carboxyl groups [74]. The O 1s peak at 533.23 eV is due to hydroxide, adsorbed oxygen and/or adsorbed water at the electrode surface [75,76]. For the O 1s, the 532.3 eV peak indicates the C-O/C=O bonds. Being higher than 532 eV indicates a higher contribution from the C-O bond than from the C=O bond. The 533.3 eV peak is due to the C-OH bond, while the 534.7 eV peak is due to water molecule oxygen [75]. This could be related to MWCNT surface-functionalized groups.

The Pt 4f peak overlaps with Pt 4d and O 1s lines, characteristic of constituents of the PtNP@MWCNT-FTO-E electrode. The Pt 4f doublet (with Pt $4f^{7/2}$ and Pt $4f^{5/2}$ lines) is observed for PtNP@MWCNT-FTO-E. The lower binding energy components, 71.2 eV for Pt $4f^{7/2}$ and 74.6 eV for Pt $4f^{5/2}$, are due to metallic Pt [77–80]. The higher binding energy components, 71.7 eV for Pt $4f^{7/2}$ and 75.0 eV for Pt $4f^{5/2}$, are due to Pt(OH)$_x$ or PtO [77–80]. The O 1s peak at 532.7 eV is due to adsorbed oxygen or water. The peak at 530.9 eV is due to adsorbed OH on the deposited Pt nanoparticles or possibly PtO [79,80].

In the XPS spectrum for the used PtNP@MWCNT-FTO-E electrode, a new peak at 200 eV due to Cl can be observed. This is attributed to Cl ions coming out from originally added phenazopyridine HC to the reaction solution. The Cl ions may adsorb onto the electrode.

The XRD and EDS results thus confirm the presence of Pt in the PtNP@FTO-T and PtNP@FTO-E electrodes. EDS and XPS spectra further confirm the presence of Pt in the fresh and used PtNP@MWCNT-FTO-E electrodes.

### 3.2. Effect of Electrode Type on Phenazopyridine Loss%

- The Pt@FTO electrodes:

Various electrodes, including naked FTO, Pt sheet, PtNP@FTO-T and PtNP@FTO-E, are examined in aqueous phenazopyridine electrooxidation at +1.6 V, intrinsic pH 3.0 and room temperature. Figure 7a,b summarize the results. The naked FTO exhibits very low efficiency, as only 5% of the contaminant is oxidized in 120 min. The Pt sheet exhibits higher contaminant loss% than the naked FTO electrode. The Pt sheet and PtNP@FTO-E show comparable loss% values. However, knowing that the platinum total mass ($2.5 \times 10^{-2}$ g) in the PtNP@FTO-E is much smaller than that in the Pt sheet (1.71 g), the actual efficiency of the PtNP@FTO-E is much higher than that of the platinum sheet. This shows the importance of using Pt nanoparticles in the electrooxidation process. The PtNP@FTO-T electrode shows lower contaminant loss% than the PtNP@FTO-E, with statistical significance. However, this shows the advantage of depositing the platinum electrically rather than thermally. In addition, the Pt mass in the thermally prepared electrode is larger than that in the electrochemically prepared electrode, as shown in Section 2.3 above. Therefore, PtNP@FTO-E has higher efficiency per gram of Pt.

- The Pt@MWCNT electrodes:

MWCNT-FTO and PtNP@MWCNT-FTO-E electrodes are assessed in phenazopyridine electrooxidation (Figure 8).

Figure 8 confirms that the presence of Pt nanoparticles has a significant impact on the phenazopyridine loss%. In spite of its lower Pt amount, as described in Section 2.3 above, the electrode PtNP@MWCNT-FTO-E induces higher phenazopyridine loss% than PtNP@FTO-E (in Figure 7). This is attributed to the spongy nature of the MWCNT matrix, as discussed in the mechanism section.

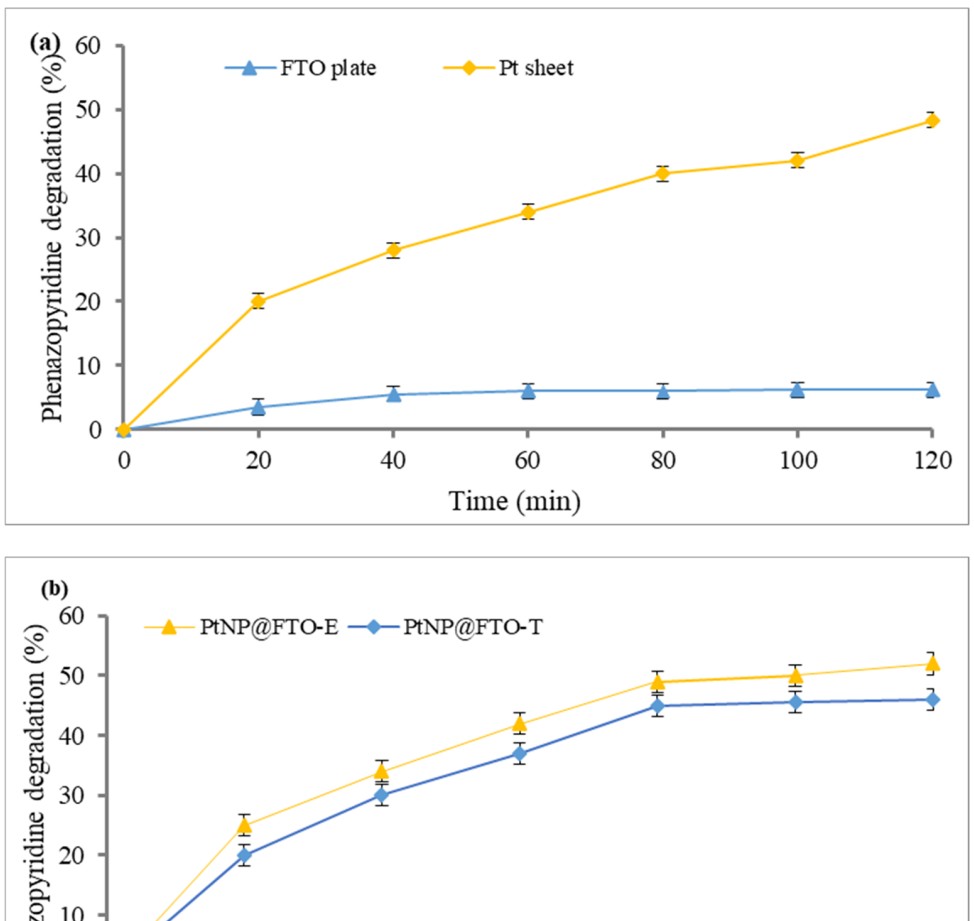

**Figure 7.** Phenazopyridine electrooxidation over time on (**a**) naked FTO and Pt sheet, (**b**) PtNP@FTO-T and PtNP@FTO- E. Experiments are conducted using phenazopyridine solutions (70 mL, 60 ppm) at +1.60 V (vs. SCE) at room temperature and intrinsic pH. The interelectrode spacing is 1.2 cm.

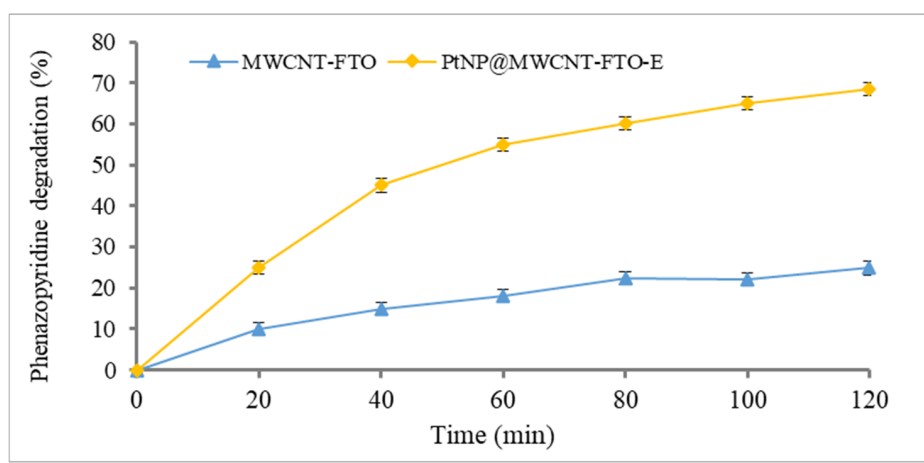

**Figure 8.** Phenazopyridine electrooxidation over time on MWCNT-FTO and PtNP@MWCNT-FTO-E. Experiments are conducted using phenazopyridine solution (70 mL; 40 ppm) at +1.60 V (vs. SCE) at room temperature and intrinsic pH. Interelectrode spacing is 1.2 cm.

Figures 7 and 8 indicate that both Pt nanoparticles and the MWCNT matrix synergistically increase the phenazopyridine loss% in the electrooxidation process. For these reasons, the novel PtNP@MWCNT-FTO-E electrode is singled out here to conduct more investigations. The effects of reaction conditions on phenazopyridine loss% are described here.

- Cyclic voltammetry

Results of electrode type effect on phenazopyridine degradation are further confirmed by cyclovoltammetry (CV). Cyclo-voltammograms are measured for the electrodegradation reaction progress using all electrodes, Pt sheet, naked FTO, PtNP@FTO and PtNP@MWCNT-FTO-E. Comparison between various electrodes, under otherwise same conditions, is made in terms of peak position and peak height. The literature shows that pH may affect peak position and height in the voltammogram [81]. Even at the same pH, the buffer type may affect the peak position [82]. Moreover, the scanning rate affects the peak height [81] in the voltammogram. Therefore, all these conditions are kept constant in the CV comparative study here.

Figure 9 shows voltammograms for the examined electrodes. Peak positions for phenazopyridine can be observed at 1.0 V (vs. SCE). The oxidation peak is slightly lower than earlier described in a paper [81], with 1.1 V (vs. SCE) for phenazopyridine at pH 7. This slight difference is understandable, as the paper used a high phenazopyridine HCl concentration of 250 ppm. As phenazopyridine HCl itself behaves as an electrolyte, the total electrolyte concentration was higher than in this study. The literature shows that using higher electrolyte concentrations shifts peak positions to higher potentials [83]. Only the oxidation peak is observed, with no significant reduction, which indicates the irreversible removal of phenazopyridine.

The figure shows that at ~1.0 V (vs. SCE), phenazopyridine HCl peak height values vary for various electrodes. The results are summarized in Table 4. The table shows that the oxidation peak current varies for various electrodes in the following order: MWCNT-FTO < naked FTO < Pt sheet < PtNP@FTO-E < PtNP@MWCNT-FTO-E. The CV results are thus congruent with all results discussed above and confirm the special value of using the new electrode here.

The MWCNT-FTO electrode shows lower activity than the naked FTO, presumably due to higher resistance at the solid/liquid interface in the former. Based on that, the PtNP@MWCNT-FTO-E electrode should also have lower activity due to high resistance against MWCNTs. Despite that, the PtNP@MWCNT-FTO exhibits the highest activity. This is partly due to the increased surface area for PtNPs at the MWCNT mesh, as described above. Moreover, the PtNP coatings at the single MWCNTs may extend and have direct contact with the FTO substrate, as observed from SEM micrographs, which lowers resistance and facilitates charge transfer.

**Table 4.** Showing values of CV peak potential and peak current for phenazopyridine using various electrodes.

| Entry # | Electrode | Peak Voltage (V) vs. SCE | Peak Oxidation Current Density (mA/cm$^2$) |
|---|---|---|---|
| 1 | PtNP@MWCNT-FTO-E | 1.0 | 25 |
| 2 | Pt sheet | 1.0 | 10 |
| 3 | PtNP@FTO-E | 0.8 | 17 |
| 4 | MWCNT-FTO | 1.0 | 0.5 |
| 5 | Naked FTO | 1.0 | 1.5 |

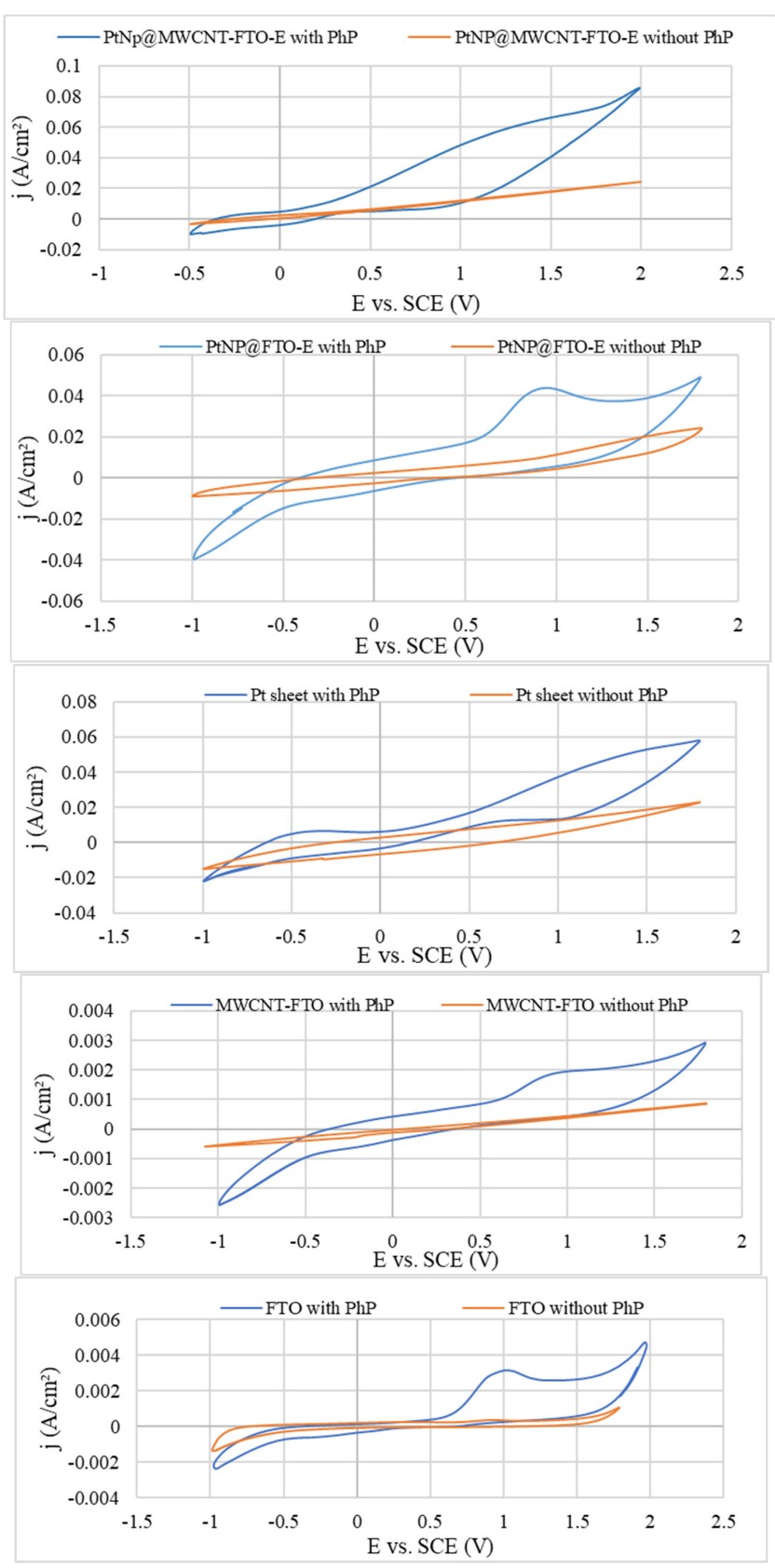

**Figure 9.** Cyclic voltammograms measured for various electrodes. All measurements are taken in buffered solution of pH 7 at scan rate 50 mV/s, 30 cycles, vs. SCE. Phenazopyridine 250 ppm in 100 mL methanol–water (30:70 *v/v*) at room temperature.

### 3.3. PtNP@MWCNT-FTO-E Electrode in Phenazopyridine Removal

3.3.1. Effect of Reaction Parameters

- Applied bias effect:

Phenazopyridine electrooxidation is studied at various applied voltages (Figure 10). The figure shows that the phenazopyridine loss is affected by the applied potential. At +1.20 V (vs. SCE), the loss% is relatively low, while at higher potential, the loss% is higher. That is due to the higher driving force associated with higher potentials. However, at 1.60 and 2.0 V, no significant difference in loss% is observed. Therefore, unless otherwise stated, the working potential is set at 1.60 V (vs. SCE). This is to avoid overpotential values for economic reasons. Another reason is to avoid the occurrence of other unnecessary oxidation processes at high potentials.

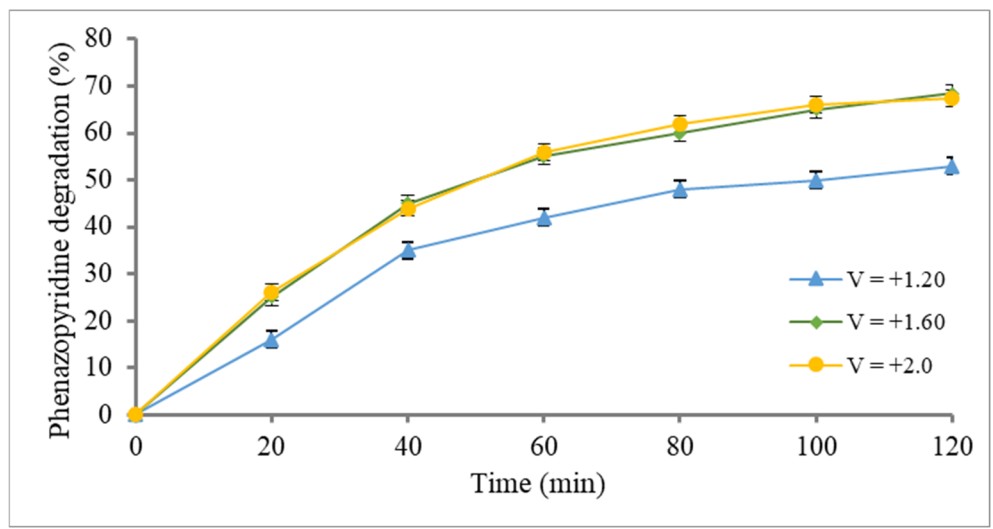

**Figure 10.** Effect of applied bias (vs. SCE) on phenazopyridine loss % using PtNP@MWCNT-FTO-E electrode. Experiments are performed in 70 mL contaminant solutions (70 mL, 40 ppm) at room temperature and intrinsic pH. The interelectrode spacing is 1.2 cm.

- Electrolyte type effect:

Phenazopyridine undergoes removal under electrooxidation conditions in the absence of any added electrolytes (Figure 11). Experiments with $Na_2SO_4$ (0.050 M) and with no added electrolyte show significantly higher contaminant loss than with NaCl (0.075 M). The added electrolyte concentrations are less than typical tap water concentrations. Therefore, these added salt concentrations pose no environmental concerns. However, the figure shows that the experiments with $Na_2SO_4$ and with no added electrolytes have no significant variation in contaminant loss%. Therefore, the present study is performed with no added electrolytes to avoid any impacts on the environment. In fact, the phenazopyridine itself includes HCl that ionizes upon dissolution.

- Temperature effect:

The effect of temperature on phenazopyridine loss% is studied using the PtNP@MWCNT-FTO-E electrode (Figure 12). The studied temperature range spans available temperature values commonly encountered in water reservoirs. The figure shows a significantly slight increase in phenazopyridine removal with increased temperature. This is attributed to increased contaminant diffusion at higher temperatures, vide the mechanism section.

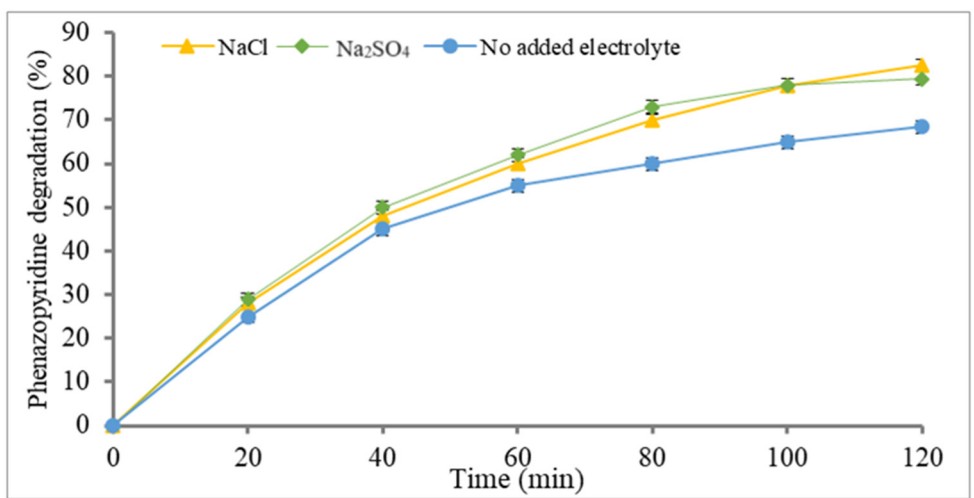

**Figure 11.** Effect of added electrolyte on phenazopyridine loss% on PtNP@MWCNT-FTO-E electrode. Experiments are conducted using solutions of phenazopyridine (70 mL, 40 ppm) at +1.60 V (vs. SCE) at room temperature and intrinsic pH with interelectrode spacing of 1.2 cm. Electrolytes are NaCl (0.075 M) or $Na_2SO_4$ (0.05 M).

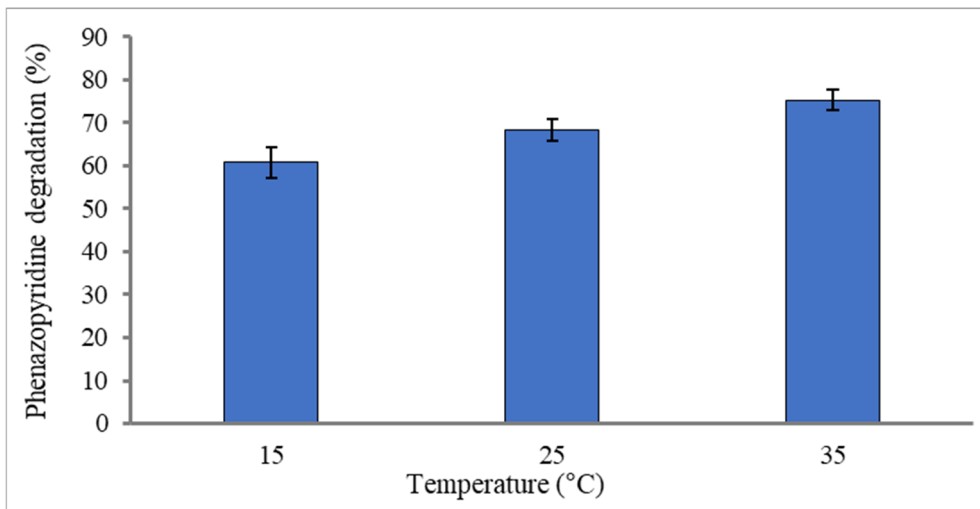

**Figure 12.** Effect of temperature on phenazopyridine degradation on PtNP@MWCNT-FTO-E electrode. Experiments are conducted using contaminant solution (70 mL; 40 ppm) at +1.60 V (vs. SCE), intrinsic pH, interelectrode spacing 1.2 cm, for 2 h.

-     Initial pH effect:

The effect of initial pH on phenazopyridine loss% is studied (Figure 13). For practical purposes, the pH is varied in the range of 3–11. Among the values, only the highly basic pH (11.0) significantly lowers the contaminant loss%. In acidic, neutral media and slightly basic (pH 9.0) media, the contaminant exhibits soundly high loss%. Thus, the PtNP@MWCNT-FTO-E electrode has the ability to electrooxidize phenazopyridine at a soundly wide pH range since fresh surface waters normally have pH in the range of 6.5–9.0, as described by the U.S. EPA [84]. In this respect, the present electrode is advantageous over earlier $IrO_2$-$Ta_2O_5$ systems [29], which function efficiently in very acidic media at pH 3.

Figure 13 shows that as pH increases, the contaminant loss% slightly increases but with no significant variation until pH 7. After that, the loss% decreases. The pH effect explanation involves two concepts with opposing effects, namely, the phenazopyridine molecular charges and the compound solubility.

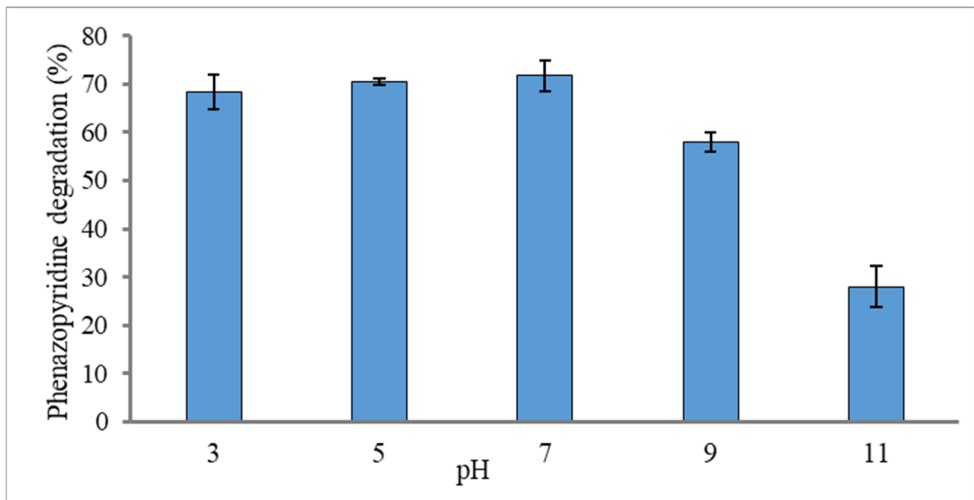

**Figure 13.** Effect of initial pH on phenazopyridine loss% using PtNP@MWCNT-FTO-E electrode. Reaction are performed at +1.60 V (vs. SCE) in contaminant solutions (70 mL, 40 ppm) at room temperature for 2 h. The interelectrode spacing is 1.2 cm.

Molecular charges: At lower pH values, the phenazopyridine molecules carry positive charges. At pH lower than 5.1, phenazopyridine is positively charged, while at higher values, it is neutral [85]. As pH increases, the molecules carry neutral charges. Thus, the molecules are more attracted to the positively charged working electrode, and consequently, the contaminant loss % is improved with increased pH.

Solubility: Phenazopyridine solubility in water increases at lower pH, specifically at pH 4. This explains why the contaminant loss% is high at lower pH values (3–7). At higher pH values, the solubility decreases sharply, as reported earlier [86].

- Initial phenazopyridine concentration effect:

The effect of variation of initial phenazopyridine concentration on its removal, at +1.60 V on the PtNP@MWCNT-FTO-E electrode, is studied (Figure 14). At higher contaminant concentrations, the loss% decreases. However, the amount of removed contaminant increases with increased concentration. This indicates the practical value of the electrode in treating waters contaminated with various phenazopyridine concentrations. The effect of the contaminant's initial concentration is further discussed in Section 3.3.2.

- Effect of reaction time:

The results described above are observed after 120 min reaction time. The contaminant removal is 60% or higher, depending on reaction conditions. A high phenazopyridine concentration (100 ppm) is intentionally used here to check the feasibility of the PtNP@MWCNT-FTO-E electrode in contaminant removal. When left for enough time, ca. 300 min, the phenazopyridine degradation proceeds to completion (Figure 15). The electrode is thus useful in the complete removal of phenazopyridine by electro-degradation.

- Phenazopyridine complete mineralization:

The phenazopyridine molecules electrooxidized on PtNP@MWCNT-FTO-E are completely mineralized, leaving no organic matter. This is confirmed by various methods, namely by measuring the resulting nitrate ions, electronic absorption spectra over time, HPLC analysis and TOC.

Figure 16 shows that the phenazopyridine absorbance (the azo group at ~430 nm) decreases over time. Moreover, the peak at ~280 nm, known for the stable phenyl groups, also decreases over time. This indicates that the phenazopyridine and its stable phenyl groups are degraded.

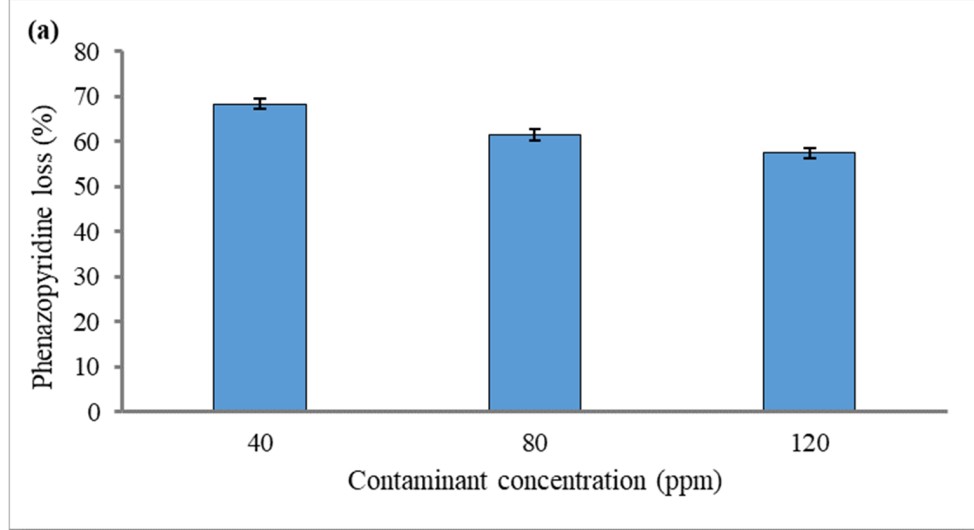

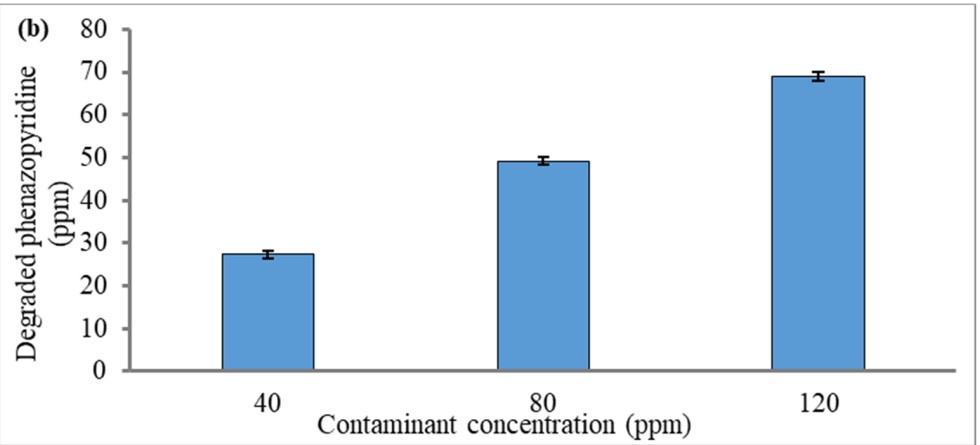

**Figure 14.** Effect of phenazopyridine concentration on its removal using PtNP@MWCNT-FTO-E electrode. (**a**) Contaminant concentration removed in ppm; (**b**) contaminant degradation%. Experiments are conducted using 70 mL solutions of various contaminant concentrations at +1.60 V (vs. SCE), room temperature and intrinsic pH for 2 h. Interelectrode spacing is 1.2 cm. Measurement error is 5%.

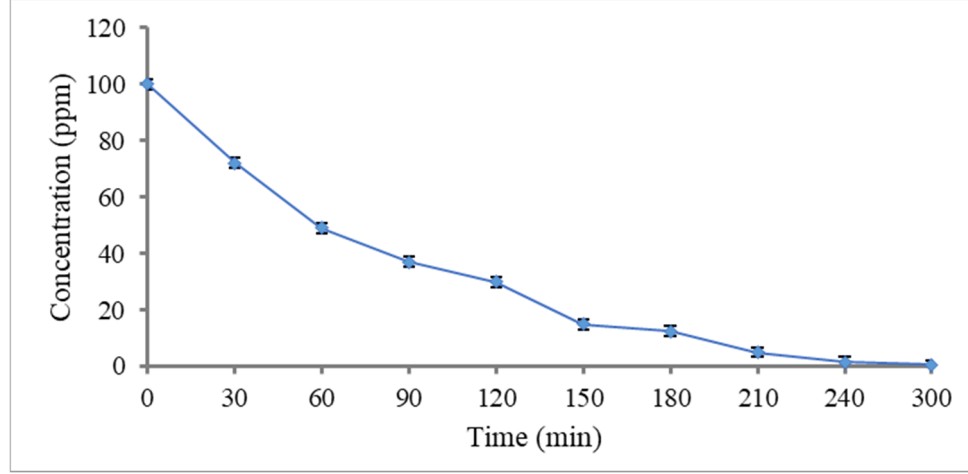

**Figure 15.** Phenazopyridine degradation over time by electrooxidation on PtNP@MWCNT-FTO-E electrode. Experiments are conducted using (70 mL; 100 ppm) at +1.60 V (vs. SCE), room temperature and intrinsic pH. Interelectrode spacing is 1.2 cm.

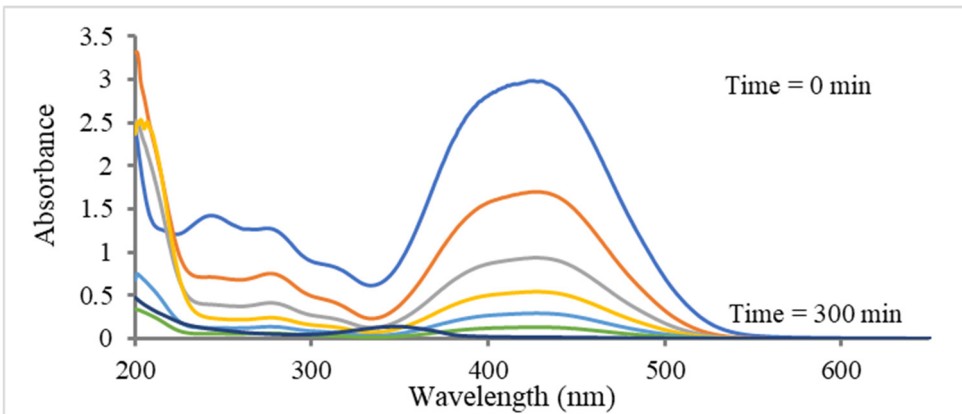

**Figure 16.** Electronic absorption spectra measured for phenazopyridine reaction solution over time. Reactions are conducted on PtNP@MWCNT-FTO-E electrode in 70 mL solutions of contaminant at 1.6 V (vs. SCE), room temperature and intrinsic pH. The interelectrode spacing is 1.2 cm.

Figure 17 shows the concentration of nitrate ion that results during the phenazopyridine electrooxidation on the PtNP@MWCNT-FTO-E electrode. The figure shows that the nitrate ion occurs during the electrooxidation process over time. This is due to oxidation of the nitrogen atoms present in the phenazopyridine molecule. However, over time, the resulting nitrate ion concentration decreases to reach very low values after 300 min. This is another evidence in favor of phenazopyridine mineralization in the process.

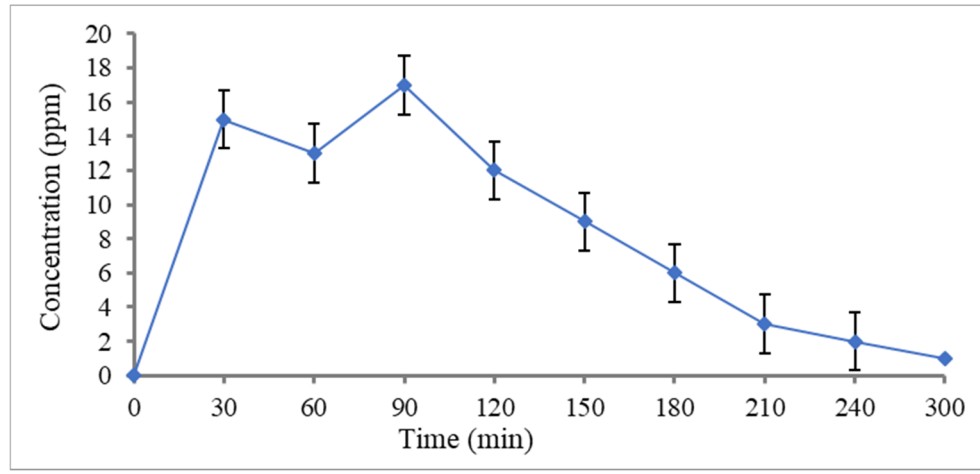

**Figure 17.** Nitrate ion production development by electrooxidation on PtNP@MWCNT-FTO-E electrode, over time. Experiments are conducted using 70 mL of highly concentrated (100 ppm) at room temperature and intrinsic pH at +1.60 V (s. SCE). Interelectrode spacing is 1.2 cm.

The TOC values, measured at various reaction times, are shown in Table 5. The organic species continue to decrease with reaction time. After 300 min, no organic compounds are observed in the reaction solution. This is another evidence in favor of complete mineralization of the phenazopyridine during the electrooxidation process. In one earlier report, organic by-products were detected in the reaction solution after phenazopyridine electrooxidation on the $IrO_2$-$Ta_2O_5$ electrode [29]. In another report, complete degradation was observed only at a low pH of 3 [33]. The present results highlight the potential value of the present electrode in future water purification processes. High-performance liquid chromatographic (HPLC) analysis also confirms the loss of phenazopyridine by electrooxidation on the PtNP@MWCNT-FTO-E electrode over time.

**Table 5.** Values of TOC and HPLC peak height measured over time for phenazopyridine electrooxidation on PtNP@MWCNT-FTO-E electrode. Experiments are conducted using contaminant solutions (70 mL, 40 ppm) at +1.60 V (vs. SCE), room temperature and intrinsic pH for 2 h. Interelectrode spacing is 1.2 cm.

| | TOC Analysis * | | HPLC Analysis | |
| --- | --- | --- | --- | --- |
| **Time (min)** | **TOC (ppm)** | **% Loss** | **HPLC (μV s)** | **% Loss** |
| 0 | 40 | 0 | 7,152,571 | 0 |
| 60 | 22 | 46 | 3,939,397 | 44 |
| 120 | 10 | 75 | 2,246,713 | 69 |
| 180 | 3 | 93 | 527,963.0 | 92 |
| 240 | <1 | 98 | 104,903.0 | 98 |
| 300 | <<1 | 99 | 4291.000 | 99 |

* Measurement error 10%.

All results thus confirm the loss of phenazopyridine and its complete mineralization of the reacted molecules by electrooxidation. This shows the practical value of the present electrode in the complete removal of phenazopyridine, leaving no organic or nitrate ions in the treated water at various pH values, as described in Figure 13 above.

- Electrode stability and reuse:

The stability of the PtNP@MWCNT-FTO-E electrode after recovery and reuse is studied in phenazopyridine electrooxidation. Figure 18 confirms that the fresh electrode loses no significant efficiency on the first or second reuse. The loss% values observed in Figure 18 are similar, with no statistical variation. The electrode stability on recovery and reuse is further confirmed by characterization results in Section 3.2 above. All XRD, SEM, EDS and XPS results indicate no change in the PtNP@MWCNT-FTO-E electrode upon reuse. The results confirm the potential of using the new electrode in phenazopyridine contaminant removal from water as a stable and safe electrode, leaving no potentially hazardous organics or minerals after treatment.

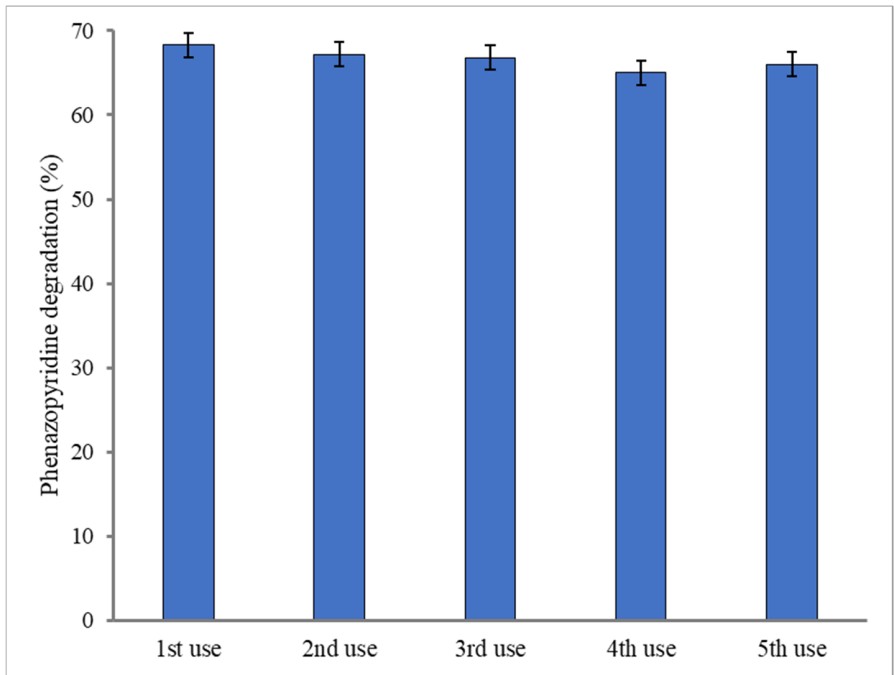

**Figure 18.** PtNP@MWCNT-FTO-E electrode stability on recovery and reuse in phenazopyridine electrooxidation. Experiments are conducted using phenazopyridine solutions (70 mL, 40 ppm) at +1.60 V (vs. SCE), intrinsic pH and room temperature for 2 h. Interplanar spacing is 1.2 cm.

### 3.3.2. Kinetics of Phenazopyridine Electrooxidation

The kinetics for phenazopyridine electrooxidation on the PtNP@MWCNT-FTO-E electrode are studied using the two known methods, namely the integrated rate law [87] and the initial rate law [88] methods. A detailed description of the kinetic study is outlined in the Supplementary Materials.

For the integrated rate method (Figure S3), the squared correlation coefficient ($R^2$) values for zero-, first- and second-order models are 0.9266, 0.9972 and 0.9270, respectively. The $R^2$ value for the first-order model is slightly higher than other models, which indicates that the first order is more suitable to describe the phenazopyridine electrooxidation process at the PtNP@MWCNT-FTO-E electrode. The literature also shows that the phenazopyridine electrooxidation on other electrodes follows nearly first-order kinetics [13].

The integrated rate order method is not conclusive, as it does not provide the reaction order value. The method of initial rate law [88] is examined. Three electrooxidation experiments, with various initial concentrations of phenazopyridine (40 ppm, 80 ppm and 120 ppm), are conducted. The general formula of the reaction rate is described in the Supplementary Materials based on the literature [89,90].

Based on the initial rate method (Figure S4), the order of the reaction, with respect to phenazopyridine, is 0.69. The rate constant is $8.35 \times 10^{-2}$ min$^{-1}$. It should be noted that the kinetics are measured at the intrinsic pH value of 3.0. At pH 7.0, the reaction is faster, and the rate constant should be higher than this value.

The literature also showed that the phenazopyridine electrooxidation obeyed the pseudo-first order [32,91–94]. The rate constant value is, therefore, preferentially obtained from the initial rate method as $8.35 \times 10^{-2}$ min$^{-1}$. Values of n < 1 reveal that phenazopyridine is adsorbed, together with other species, at the electrode surface [21,28,32].

Reports describing electro-degradation of phenazopyridine [29,32,33] do not show values of reaction rates. Therefore, the present results could not be quantitatively compared with these reports. However, in the case of magnetite@activated carbon [33], the reaction goes faster only at lower phenazopyridine concentrations at low pH values. The Fe-doped ZIF8 electrode is effective at neutral pH values, only very low phenazopyridine concentrations [32]. The Ta-based $IrO_2$-$Ta_2O_5$ electrode efficiently degrades phenazopyridine at pH 8 and relatively high concentrations [29].

As per environmental impact, the PtNP@MWCNT-FTO-E poses no threat to the environment. This is due to Pt chemical stability during electrooxidation. The Pt metal itself has no known toxicity or environmental impact, as reported in Fisher MSDS.

The PtNP@MWCNT-FTO-E electrode involves nanoparticles of Pt. This drastically lowers the processing costs. In Pt sheets, electrodes of $2 \times 2$ cm$^2$ involving 1–2 g are needed. The present electrode involves $1.0 \times 10^{-2}$ g Pt distributed over an 8 cm$^2$ area. Moreover, the higher activity of the new electrode, compared to the Pt sheet, adds to its economic advantages.

### 3.3.3. A Plausible Mechanism:

The results confirm the complete aqueous phenazopyridine degradation by electrooxidation reaction on the PtNP@MWCNT-FTO-E electrode. The electrooxidation process involves Pt, phenazopyridine, solvent water and other possible materials in addition to electrons.

As per Pt, mechanisms of electrooxidation of specifically phenazopyridine are not proposed. Mechanisms of Pt electrooxidation of other compounds, such as ethanol [95,96], glycerol [95], methanol [97], formic acid [98] and others, are described. The phenazopyridine electro-degradation mechanisms on other metal electrodes, such as $IrO_2$-$Ta_2O_5$, are reported [29], as well as magnetite/carbon [33]. Phenazopyridine is also degraded by ozone oxidation and iron compounds [28]. The mechanisms are based on the formation of the short-lived active species, mainly hydroxyl radical (OH$^.$), which are responsible for the complete degradation of the contaminant molecules [99].

Electrooxidation mechanisms of various other waste pharmaceuticals were also proposed earlier. In some cases, the processes rely solely on short-lived active species OH· radical [100]. However, another mechanism via direct oxidation of the contaminant molecules at the electrode surface cannot be ruled out [101–103].

Based on earlier literature, a general two-route mechanism is proposed here for the electro-reduction of phenazopyridine on the PtNP@MWCNT-FTO-E electrode, as described in Scheme 2.

**Scheme 2.** A plausible mechanism for phenazopyridine electrooxidation. Two possible routes are shown.

The proposed mechanism is not new by any means but is rather a plausible one based on the literature described above. The mechanism successfully explains all observations found in the present work, as follows:

- Platinum nanoparticles have higher efficiency than Pt sheets. This is due to the fact that the nanoparticles have a much higher specific surface area than the bulk Pt sheet. This facilitates the charge transfer between the Pt particle surface and other species;
- Pt nanoparticles in PtNP@FTO-E are more efficient than naked MWCNTs. This is rationalized by the higher known conductivity for Pt compared to MWCNTs;
- PtNP@MWCNT-FTO-E is more efficient than PtNP@FTO-E. This is rationalized as follows. In the former electrode, the Pt particles are randomly distributed on the MWCNT surface as isolated particles in a diffused fashion. In the latter electrode, the Pt particles exist inside a packed multilayer film. Therefore, in the former electrode, water and contaminant molecules can penetrate and reach more Pt particles than in the compact film. Logically, in the latter electrode, only upper-layer particles are accessible to molecules. This is further corroborated by the effect of temperature on reaction progress. At higher temperatures, the PtNP@MWCNT-FTO-E activity increases due to increased molecular diffusion;
- The increase in reaction speed at higher contaminant concentration in the case of PtNP@MWCNT-FTO-E is also rationalized by the increased penetration of the contaminant inside the MWCNT matrix;
- As described above, the effect of pH on reaction rate can be explained by two opposing factors: *charge and solubility*. With increased pH, the phenazopyridine molecules carry less positive charge and become more attracted to the positive electrode. In either route of the proposed mechanism, the ability of the molecules to live in close proximity to the electrode is a condition for electrooxidation. In the case of Route A, the short-lived species (hydroxyl radicals) live for a very short time near the electrode surface. Therefore, the contaminant molecules must be close to the electrode to be affected. Similarly, in Route B, the contaminant molecules must be closer to the electrode surface to be affected. As the pH increases, the contaminant becomes less soluble with less accessibility to the electrode, and consequently, the reaction slows down;

- Based on the pH discussions above, adsorption of phenazopyridine onto the Pt particles is encouraged at nearly neutral pH. With high SSA values, the particles may behave as charge transfer catalysts at the solid/liquid interface.

All in all, this study proposes a new efficient and safe electrode for the electro-mineralization of aqueous pharmaceutical contaminants, such as phenazopyridine. The present study is performed at the lab scale only. While the process shows promising findings as described above, more investigation is still needed to scale up the process at the pilot plant scale. That is to examine the technical, financial and environmental feasibility at the commercial scale. Moreover, examining the process in a continuous-flow reaction system is worth studying in the future. Work is underway in these laboratories to achieve these objectives. The study also opens a new research area for water purification from pharmaceuticals and other organic contaminants.

## 4. Conclusions

Platinum nanoparticles are deposited onto FTO/Glass substrates by thermal spray and by electrochemical deposition. The nanoparticles are also electrodeposited onto multi-walled carbon nanotubes that are pre-deposited onto FTO/Glass substrates. All resulting Pt nanoparticle electrodes are characterized and used in the electro-degradation of phenazopyridine from water. The Pt nanoparticles exhibit higher efficiency in the electro-degradation process than the Pt sheet or the naked FTO/Glass electrodes. The inclusion of the MWCNT matrix further improves the efficiency of the electrodes with Pt nanoparticles. Due to its high efficiency, the PtNP@MWCNT-FTO-E electrode is studied more extensively. The electrode is stable upon recovery and reuse, showing no signs of efficiency loss or change in XRD, SEM, EDS or XPS properties. The electrode completely removes the phenazopyridine from water, leaving no organic materials, as confirmed by spectra, nitrate ion measurement, TOC and HPLC measurements. The reaction order with respect to the contaminant is 0.69, which means the electrode is more efficient with increased concentration. The new electrode has the potential to be used in future large-scale water purification processes, with phenazopyridine as a model pharmaceutical contaminant.

**Supplementary Materials:** The following supporting information can be downloaded at https://www.mdpi.com/article/10.3390/pr12081625/s1.

**Author Contributions:** I.M.N.: Investigation (preparations and reactions), writing—original draft. H.N.: Validation, training, supervision, writing—review and editing. M.A.: Supervision, resources, validation, writing. M.H.S.H.: Conceptualization (of studying pharmaceuticals), validation, writing—review and editing. T.W.K.: Supervision, investigation, characterization, H.H.: Investigation, solid material characterization. M.S.: Investigation, TOC measurement, statistical analysis. H.S.H.: Supervision, project administration, writing—review and editing. All authors have read and agreed to the published version of the manuscript.

**Funding:** T.W.K. acknowledges support from "the National Research Council of Science & Technology (NST) grant by the Korean government (MSIT) (No. CAP20034-200)". Financial support for the thesis from ANU, based on the thesis funding policy, is acknowledged. No special funding is donated.

**Data Availability Statement:** Data will be available upon request.

**Acknowledgments:** The results were acquired by I.N. as part of his Ph.D. thesis at ANU, Palestine. The authors wish to thank Ameed Amireh, of the technical staff at ANU, for their continued help and devotion. I.N. acknowledges technical help from Moath Omer of Palestine Technical University, Toulkarem.

**Conflicts of Interest:** The authors declare no conflicts of interest.

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
