# Peer review of "Electro-Mineralization of Aqueous Phenazopyridine Using Platinum Nanoparticles Deposited onto Multiwalled Carbon Nanotubes"

_processes, doi:10.3390/pr12081625_

Round 1
Reviewer 1 Report
Comments and Suggestions for Authors
This manuscript gives the synthesis of carbon nanotubes supported Pt particles and its electrocatalytic performance. The structure, composition and morphology of hybrids were characterized, and their phenazopyridine electrooxidation performances were demonstrated. This manuscript appeared to lack of novelty. Moreover, it should be signified that a more systematic work should be performed to clarify the "phenazopyridine electrooxidation", for instance the cyclic voltammogram tested curve. Despite much work finished, the manuscript shows insufficient systematic research and not strong logic. It is a suggestion to check the methodology of the entire electrochemical experimental part. The performance comparison of the reported Pt-based electrodes should be added to prove the evident competition superiority.
Comments on the Quality of English LanguageThe English language in this article is fluent without much errors.
Author Response
- This manuscript gives the synthesis of carbon nanotubes supported Pt particles and its electrocatalytic performance. The structure, composition and morphology of hybrids were characterized, and their phenazopyridine electrooxidation performances were demonstrated.
Response: Thanks to our Reviewer for patience and advice.
- This manuscript appeared to lack of novelty.
Response: Again, thanks to our reviewer. Novelty of this work has now been rewritten and highlighted at the bottom of the Introduction in color highlight. Please note that the prepared electrodes PtNP@MWCNT-FTO-T and PtNP@MWCNT-FTO-E are all novel and were not described before. The electrodes are also used here in electro-mineralization of contaminant pharmaceuticals (phenazopyridine) for the first time.
We hope that novelty is now better highlighted
Please see all yellow and green highlights.
- Moreover, it should be signified that a more systematic work should be performed to clarify the "phenazopyridine electrooxidation", for instance the cyclic voltammogram tested curve.
Response: Thanks to our reviewer.
More work has been added. Please see color highlights (in yellow and green).
Moreover, CV study has been performed to confirm phenazopyridine removal. Please see Discussions of the new Figure 8 and Table 4 for details.
- Despite much work finished, the manuscript shows insufficient systematic research and not strong logic. It is a suggestion to check the methodology of the entire electrochemical experimental part.
Response: Thanks to our reviewer. These comments have now been considered. We hope that the revised manuscript (with highlights) satisfies all concerns pointed out by the reviewer.
Please see also the new discussions on cyclic voltammetry.
- The performance comparison of the reported Pt-based electrodes should be added to prove the evident competition superiority.
Response: If our reviewer means by “reported Pt-based electrodes” as “literature ones” we cannot show that because literature does not show Pt electrodes in pharmaceutical or phenazopyridine mineralization.
However, if “reported Pt-based electrodes” means from our work, we have now included a thorough comparison, as described in Sections 3.
Comments on the Quality of English Language
The English language in this article is fluent without much errors.
Response: We thank our reviewer. In addition to this comment, we have thoroughly revised the English and writing clarity.

Reviewer 2 Report
Comments and Suggestions for Authors
The manuscript presents a detailed study on the electrochemical oxidation of phenazopyridine using a following electrodes: Pt Sheet Electrode, FTO Electrode, PtNP@FTO-T, PtNP@FTO-E, MWCNT-FTO, PtNP@MWCNT-FTO-E. The authors demonstrate the preparation, characterization, and application of those electrodes for the mineralization of phenazopyridine, a common pharmaceutical contaminant in water.
Strengths:
The study introduces a novel electrode configuration that combines the high surface area and conductive properties of MWCNTs with the catalytic efficiency of Pt. This combination appears to significantly enhance the electrochemical degradation efficiency of phenazopyridine.
The authors utilize various advanced techniques such as XRD, SEM, EDS, and XPS to thoroughly characterize the electrode, confirming the successful deposition of Pt and the stability of the electrode before and after use.
The manuscript provides a plausible mechanism for the electrooxidation process, supported by experimental data and literature references. The proposed two-route mechanism involving both direct oxidation at the electrode surface and indirect oxidation via hydroxyl radicals is well-supported.
The study addresses an important environmental issue by focusing on the removal of phenazopyridine, a persistent pharmaceutical contaminant, from water. The findings have potential implications for large-scale water purification processes.
Areas for Improvement:
Cost Analysis of Pt Usage: While the authors acknowledge the high cost of Pt and propose the use of nanoparticles to reduce the overall Pt content, a more detailed cost analysis would strengthen the argument for the economic feasibility of this approach. The potential for scaling up and cost implications should be discussed.
Cross-Sectional Analysis (Figure 3): There is an inconsistency in Figure 3a2 where the cross-section appears to be that of the FTO electrode rather than the intended sample. This needs to be corrected to accurately represent the morphology and thickness of the deposited layers.
Professional Presentation of Data (Figure 5): The data in Figure 5 are presented as screenshots from software, which lacks professionalism. The figures should be recreated using appropriate graphing tools to enhance clarity and presentation quality.
Comparative Analysis in Figures 6 and 7: Figures 6 and 7 present data for fresh and used PtNP@MWCNT-FTO-E electrodes. To better illustrate differences, the spectra before and after use should be presented on the same graph. Similarly, comparative analyses of FTO, Pt, and PtNP@FTO electrodes should be consolidated into single figures to improve readability.
Statistical Analysis of Contaminant Loss (Figure 7): The authors note a slight difference in contaminant loss between PtNP@FTO-T and PtNP@FTO-E electrodes. However, the potential for measurement error is not sufficiently addressed. Statistical analysis should be included to validate the significance of the observed differences.
Inclusion of Additional Data (Figure 8): Adding data for pure Pt or PtNP@FTO-E in Figure 8 would provide a clearer comparison and better illustrate the performance enhancement achieved by incorporating MWCNTs.
Surface Area Measurement: The study would benefit from direct measurement of the active surface area of the electrodes. This could provide further evidence to support the authors' claim that the spongy nature of the MWCNT matrix contributes to higher contaminant loss.
Voltammetric Analysis: Extending the study to include comparative voltammetric analysis of all prepared electrodes could offer deeper insights into the impact of MWCNTs on the redox potentials of phenazopyridine, enhancing the overall understanding of the electrooxidation process.
Correlation Coefficient Values: The manuscript mentions negative R² values, likely referring to Pearson's “r” coefficents. This should be corrected to accurately reflect the statistical analysis performed.
Nanoparticle Confirmation: While the Scherrer equation applied to XRD patterns suggests the presence of nanoparticles, this alone is not sufficient to conclusively confirm their existence. It is recommended to include additional characterization techniques such as Transmission Electron Microscopy (TEM) or Atomic Force Microscopy (AFM) to directly visualize the nanoparticles and confirm their size, shape, and distribution. This would provide more robust evidence supporting the claim of nanoparticle formation. Moreover, considering the SEM images presented in the manuscript, the deposited platinum layer appears to be relatively thick. This observation suggests that it may be challenging to assert that the Pt is present solely as nanoparticles. The thickness and uniformity of the Pt layer observed in SEM images indicate that the deposition result in larger aggregates or continuous films rather than discrete nanoparticles. Therefore, integrating TEM or AFM analyses would be crucial to accurately determine the presence and nature of the nanoparticles, if they exist.
Long-term Stability Studies: Conduct and report on the electrode's performance over an extended number of cycles.
Comparative Analysis: Compare the new electrodes with existing technologies regarding efficiency, cost, and environmental impact.
Overall, the manuscript provides significant contributions to the field of environmental electrochemistry, presenting a novel and efficient approach for the degradation of phenazopyridine. Addressing the areas for improvement will enhance the clarity, accuracy, and impact of the study, making it a valuable resource for future research in water purification technologies.
Comments on the Quality of English Language
"electron-phenton process" line 52 should be Electro-Fenton Process
The text contains several ambiguities that may affect its clarity and readability. A professional language review would significantly improve the overall quality and precision of the document. Few exaples below:
"While using Pt electrodes can be useful, their high cost is a limitation..." line 70; I assume this refers to the total cost of Pt; the use of nanoparticles significantly reduces the demand for Pt.
"However, reports describing removal of drugs and phenazopyridine from water using Pt based electrodes, let alone nanoscale Pt (PtNP), are not widely available" line 66-68; I assume the authors meant there is a limited number of studies on this topic.
Author Response
Comments and Suggestions for Authors
The manuscript presents a detailed study on the electrochemical oxidation of phenazopyridine using a following electrodes: Pt Sheet Electrode, FTO Electrode, PtNP@FTO-T, PtNP@FTO-E, MWCNT-FTO, PtNP@MWCNT-FTO-E. The authors demonstrate the preparation, characterization, and application of those electrodes for the mineralization of phenazopyridine, a common pharmaceutical contaminant in water.
Strengths:
The study introduces a novel electrode configuration that combines the high surface area and conductive properties of MWCNTs with the catalytic efficiency of Pt. This combination appears to significantly enhance the electrochemical degradation efficiency of phenazopyridine.
The authors utilize various advanced techniques such as XRD, SEM, EDS, and XPS to thoroughly characterize the electrode, confirming the successful deposition of Pt and the stability of the electrode before and after use.
The manuscript provides a plausible mechanism for the electrooxidation process, supported by experimental data and literature references. The proposed two-route mechanism involving both direct oxidation at the electrode surface and indirect oxidation via hydroxyl radicals is well-supported.
The study addresses an important environmental issue by focusing on the removal of phenazopyridine, a persistent pharmaceutical contaminant, from water. The findings have potential implications for large-scale water purification processes.
Response: We value our reviewer’s comments
Areas for Improvement:
Cost Analysis of Pt Usage: While the authors acknowledge the high cost of Pt and propose the use of nanoparticles to reduce the overall Pt content, a more detailed cost analysis would strengthen the argument for the economic feasibility of this approach. The potential for scaling up and cost implications should be discussed.
Response: As described in green highlight in the Introduction, using Pt nanoparticles is widely considered in applied and applied research, and in industrial process. The reasoning is presented in the Introduction Section.
As per the potential to scale up our process, we have not conducted any pilot plant study in the area. These issues are now discussed at the end of Results and discussion section (just above Conclusions in green highlights).
We hope our reviewer comments are now met. If not, we will be ready to re-address all comments again.
Cross-Sectional Analysis (Figure 3): There is an inconsistency in Figure 3a2 where the cross-section appears to be that of the FTO electrode rather than the intended sample. This needs to be corrected to accurately represent the morphology and thickness of the deposited layers.
Response: Thanks to our reviewer. The cross-sectional images have been replaced with clearer images from our collaborators in Korea. The discussions have now been revised in the discussions for Figure 3 and 4. All highlighted in green highlight, in the section Surface Morphology.
Professional Presentation of Data (Figure 5): The data in Figure 5 are presented as screenshots from software, which lacks professionalism. The figures should be recreated using appropriate graphing tools to enhance clarity and presentation quality.
Response: We agree with our reviewer. The Figure was taken from the source as it is, noneditable. We kept it as is and we do not wish to manipulate for scientific authenticity. However, it has now been shifted to the supplementary information as Figure S1. The values are presented in the new Table 3 in the manuscript.
Please note that other Figure and Table numbers have now changed accordingly.
Comparative Analysis in Figures 6 and 7: Figures 6 and 7 present data for fresh and used PtNP@MWCNT-FTO-E electrodes. To better illustrate differences, the spectra before and after use should be presented on the same graph. Similarly, comparative analyses of FTO, Pt, and PtNP@FTO electrodes should be consolidated into single figures to improve readability.
Response: I hope that we understood what our reviewer demands. Please note changes in Figure numbers.
- As our reviewer demands, the new Figure 5 shows fresh and used in a single figure
- The new Figure 6 describes all Pt, FTO, PtNP@FTO-T and PtNO@FTO-E electrodes
- The new Figure 7 describes MWCNT-FTO and PtNP@MWCNT-E for comparison and clear discussion.
- Please note we do not wish to overload figures with data to make things easily understandable.
Statistical Analysis of Contaminant Loss (Figure 7à6): The authors note a slight difference in contaminant loss between PtNP@FTO-T and PtNP@FTO-E electrodes. However, the potential for measurement error is not sufficiently addressed. Statistical analysis should be included to validate the significance of the observed differences.
Response: Thanks to our reviewer. All data have now been statistically analyzed. Please see the new Section 2.6 and other statistical data in all other Figures 6-16.
Please note the changes in the new Figure numbers.
Cyclic voltammetry could not be measured for the PtNP@FTO-T electrode, because new preparations need time which is not available.
I hope our reviewer considers our limitations in this respect.
Inclusion of Additional Data (Figure 8): Adding data for pure Pt or PtNP@FTO-E in Figure 8 would provide a clearer comparison and better illustrate the performance enhancement achieved by incorporating MWCNTs.
Response: Thanks to our reviewer. We have combined figures as demanded by our reviewer above.
In the present Figures, we have grouped graphs and compared them based on relevance.
For instance: Pt sheet and FTO together (Figure 6a), Pt@FTO-T and Pt@FTO-E in together (Figure 6b).
MWCNT-FTO and Pt@MWCNT-FTO-E together (Figure 7)
Based on this order, the discussions have now made, hope they are clarified now.
Please note that: We cannot repeat any plot (Pt or PtNP@FTO) in two Figures. We cannot also combine all electrodes in one Figure to avoid overcrowding. But, if our reviewer and our editor insist on that, we will the do what you need.
Surface Area Measurement: The study would benefit from direct measurement of the active surface area of the electrodes. This could provide further evidence to support the authors' claim that the spongy nature of the MWCNT matrix contributes to higher contaminant loss.
Response: Thanks to our reviewer. We have measured the SSA values for various electrodes. We used the acetic adsorption method which is only approximate method, as we have no access to advanced methods. The PtNP@FTO electrode has SSA about 50 m2/g for the Pt film.
For MWCNT it has very high SSA of ~300 m2/g based on vendor specifications. For PtNP@MWCNT-FTO-E electrode the SSA is about 100 m2/g, which is lower than for MWCNT. This lowering is due to attachment of Pt particles on the MWCNT, that is also confirmed by SEM and XRD. Please see green highlights in the Characterization Section.
Voltammetric Analysis: Extending the study to include comparative voltammetric analysis of all prepared electrodes could offer deeper insights into the impact of MWCNTs on the redox potentials of phenazopyridine, enhancing the overall understanding of the electrooxidation process.
Response: Thanks to our reviewer. CV has been measured and discussed. Comparison of various electrodes has now been made (please see new discussions on the new Figure 8 and the new Table 4) in color highlight.
Please forgive our limitations as one electrode needs preparation and could not be examined (see our response above).
Correlation Coefficient Values: The manuscript mentions negative R² values, likely referring to Pearson's “r” coefficents. This should be corrected to accurately reflect the statistical analysis performed.
Response: Thanks to our reviewer. The error has now been corrected. R2 cannot be negative. Please look at the revised R2 values in Section 3.3.1 in green highlight. Please see also revisions on the supplementary materials.
Nanoparticle Confirmation: While the Scherrer equation applied to XRD patterns suggests the presence of nanoparticles, this alone is not sufficient to conclusively confirm their existence. It is recommended to include additional characterization techniques such as Transmission Electron Microscopy (TEM) or Atomic Force Microscopy (AFM) to directly visualize the nanoparticles and confirm their size, shape, and distribution. This would provide more robust evidence supporting the claim of nanoparticle formation. Moreover, considering the SEM images presented in the manuscript, the deposited platinum layer appears to be relatively thick. This observation suggests that it may be challenging to assert that the Pt is present solely as nanoparticles. The thickness and uniformity of the Pt layer observed in SEM images indicate that the deposition result in larger aggregates or continuous films rather than discrete nanoparticles. Therefore, integrating TEM or AFM analyses would be crucial to accurately determine the presence and nature of the nanoparticles, if they exist.
Response: We value our reviewer’s comments. Please note that:
- Based on tremendous literature, even in films, XRD is widely accepted as a strong proof of nano-crystallites. We normally observe agglomerates (by SEM). The agglomerates themselves involve nanoscale crystallites (from XRD). Please see our XRD discussions and new confirming references therein.
- 2) In our case, we benefited from our reviewer’s comments. The continuous films are clearly observed on the PtNP@FTO-T and PtNP@FTO-E electrodes, from SEM. However, even within the films, the nanoscale nature is confirmed not only by Scherrer, but by the very nature of the XRD patterns in the Figures. As widely confirmed in earlier literature also, short broad signals for Pt indicate nanoscale (even within the continuous films)
- In case of PtNP@MWCNT-FTO-E electrode, the Pt nanoparticles exist in a dispersed manner over the MWCNT mesh. The Pt small crystallites coat the tube surfaces of the MWCNTs, but do not form a continuous separate film all over. This is confirmed by SEM which shows that the tubes in PtNP@MWCNT-FTO-E have larger radii than the MWCNT-FTO only.
Please read our color highlighted discussions.
- The dispersed nature of Pt particles on MWCNT has now been further confirmed by AFM as demanded by our reviewer. The AFM has been measured for the PtNP@MWCNT-FTO-E and the background MWCNT-FTO. The object is now achieved. Other electrodes (which are clearly films) could not be tested because my student cannot prepare in time (especially under our exceptional circumstances and terrible conditions in Palestine. It was heroic effort from the student to achieve the revisions as it is difficult even to reach the lab.
- As per TEM: We tried our best to get the measurements in time. However, due to our limited resources, we need to send samples abroad. This needs months to get the results back. Please be considerate on this issue.
We have now clarified these points based on our reviewer’s comments. Please see the highlighted revisions in the SEM discussions. See also revisions in the EDS discussions.
However, our new discussions on film nature for PtNP@FTO electrodes are convincing, we hope. In the PtNP@MWCNT-FTO-E electrode, the nano-crystallite existence on separate CNTs, is also convincing I hope.
Long-term Stability Studies: Conduct and report on the electrode's performance over an extended number of cycles.
Response: This comment has now been met. The following revisions have now been made:
- The new Figure 17 now shows 5 uses for the electrode, with no statistically significant differences. Please see Figure 16.
Comparative Analysis: Compare the new electrodes with existing technologies regarding efficiency, cost, and environmental impact.
Response: As per efficiency, earlier reports (on electro-degradation of phenazopyridine) did not show reaction rate constants, which made quantitative comparison not easy. However, comparison has been added to the revised manuscript in green color at the end of Section 3.3.1.
As per cost and environmental impact, please see the discussions at the end of Section 3.3.1. in green highlights
Overall, the manuscript provides significant contributions to the field of environmental electrochemistry, presenting a novel and efficient approach for the degradation of phenazopyridine. Addressing the areas for improvement will enhance the clarity, accuracy, and impact of the study, making it a valuable resource for future research in water purification technologies.
Response: We surely benefited from our reviewers’ comments and we hope our manuscript is now more convincing.
Comments on the Quality of English Language
"electron-phenton process" line 52 should be Electro-Fenton Process
Response: Thanks to our reviewer. This has been corrected in two places (in green highlight) in the Introduction.
The text contains several ambiguities that may affect its clarity and readability. A professional language review would significantly improve the overall quality and precision of the document.
Response: The manuscript (all of it) has been carefully revised for English and clarity. Please see color highlights (in yellow and green).
Few examples below:
"While using Pt electrodes can be useful, their high cost is a limitation..." line 70; I assume this refers to the total cost of Pt; the use of nanoparticles significantly reduces the demand for Pt.
Response: That is true. Pt is costly, especially when Pt sheets are used. In the nanoscale, the cost goes down.
Please see green highlights on Page 3.
"However, reports describing removal of drugs and phenazopyridine from water using Pt based electrodes, let alone nanoscale Pt (PtNP), are not widely available" line 66-68; I assume the authors meant there is a limited number of studies on this topic.
Response: We could not find Pt sheets used in pharmaceutical (or phenazopyridine) electro-degradation. We also could not find reports describing Pt nanoparticles for these studies. We tried our best to find but we could not.
If our reviewer can help us find any reference about Pt in electro-degradation of pharmaceuticals and phenazopyridine, we will be grateful and we will make amendment.

Round 2
Reviewer 1 Report
Comments and Suggestions for Authors
The authors do not seem to have given a perfect answer to comment about the novelty, such as the homogeneity of the samples in terms of morphology, properties, etc., but it can be inferred from the results of the various tests that the homogeneity of the samples seems to be OK. While the electrochemical performance tests were supplemented, and it is hoped that the authors will improve the characterisation and performance testing methods in their subsequent research. Moreover, figure 11 seems missing.
Therefore, the paper should be revised.
Comments on the Quality of English LanguageThe language in this article is fluent with few grammar errors.
Author Response
Comments and Suggestions for Authors
The authors do not seem to have given a perfect answer to comment about the novelty, such as the homogeneity of the samples in terms of morphology, properties, etc., but it can be inferred from the results of the various tests that the homogeneity of the samples seems to be OK.
Response: In our earlier revisions, we have made convincing answers to our earlier reviewer’s comment “This manuscript appeared to lack of novelty.“ We highlighted our novelty in the Abstract and the Introduction, as normally done. We showed that the electrodes are novel, useful and efficient and have not been reported earlier to our knowledge. We believe that is how to state novelty.
As per sample homogeneity, morphology, properties, etc. We cannot discuss novelty in these terms because our electrode were not earlier reported.
Again, we hope our reviewer is happy with our revisions, as they seem to be OK.
While the electrochemical performance tests were supplemented, and it is hoped that the authors will improve the characterisation and performance testing methods in their subsequent research.
Response: Thanks to our reviewer. Many tests were made. New experiments were performed.
As per our subsequent research, we will make sure more characterizations will be made. More flow rate and pilot plant study will be made to test feasibility.
Moreover, figure 11 seems missing.
Response: Thanks to our reviewer. Figure 11 is missing in the first re-submission. Because there was error in Figure numbering. The issue has now been corrected.
Therefore, the paper should be revised.
Response: The manuscript has now been carefully revised.
Comments on the Quality of English Language
The language in this article is fluent with few grammar errors.
Response: Thank to our reviewer. English has now been revised again.

Reviewer 2 Report
Comments and Suggestions for Authors
The authors have revised the manuscript according to my expectations. However, some of the charts are poorly formatted, such as axis labels overlapping or units encroaching on the numerical axis, and inconsistent formatting of the charts. I am not certain if this is due to errors made by the authors or a problem with the software I am using. Therefore, I recommend a minor revision.
Author Response
Comments and Suggestions for Authors
The authors have revised the manuscript according to my expectations. However, some of the charts are poorly formatted, such as axis labels overlapping or units encroaching on the numerical axis, and inconsistent formatting of the charts. I am not certain if this is due to errors made by the authors or a problem with the software I am using. Therefore, I recommend a minor revision.
Response: We thank our reviewer.
As per charts and Figures, they have now been revised and edited accordingly.

Round 3
Reviewer 1 Report
Comments and Suggestions for Authors
The revised paper is ready for acceptance.